# Polyubiquitin architecture editing on collided ribosomes maintains persistent RQC activity

Shota Tomomatsu[1,2], Yoshitaka Matsuo [ID][1✉], Fumiaki Ohtake [ID][3], Takuya Tomita[4], Yasushi Saeki[4,5] & Toshifumi Inada [ID][1✉]

## Abstract

In ribosome-associated quality control (RQC), K63-linked poly-ubiquitination of ribosomal protein uS10 on the stalled ribosome is crucial for recruiting the RQC-trigger (RQT) complex. However, the mechanisms governing the maintenance and recycling of poly-ubiquitin architecture on colliding ribosomes remain unclear. Here we demonstrate that two deubiquitinating enzymes (DUBs), Ubp2 and Ubp3, play key roles in editing and recycling polyubiquitin chains on yeast uS10, thereby contributing to the promotion of RQC activity. Specifically, Ubp2 eliminates K63-linked polyubiquitin chains from uS10 on the free 40S subunit for recycling, while Ubp3 predominantly cleaves K48-linked di-ubiquitin and K48/K63-mixed-linkage polyubiquitin chains from uS10 on the translating ribosomes. We further demonstrate that K48-linkage-containing ubiquitin chains on uS10 of the colliding ribosome act as a negative signal for the RQT-mediated ribosome dissociation process. Collectively, our findings provide insight into the ubiquitin code in RQC, and define positive functions of two DUBs in maintaining persistent RQC activity.

**Keywords** Ribosome; Ubiquitination; Quality Control; Deubiquitinase; Ubiquitin Code
**Subject Categories** Post-translational Modifications & Proteolysis; Translation & Protein Quality

## Introduction

Persistent ribosome stalling leads to a collision with the trailing ribosome, triggering the Ribosome-associated Quality Control (RQC) pathway. This pathway ensures the prompt clearance of the stalled ribosome and the efficient elimination of its aberrant protein product (Brandman and Hegde, 2016; Inada, 2020; Joazeiro, 2017, 2019). The distinctive inter-ribosomal interface of collided ribosomes facilitates recognition by E3 ubiquitin ligases:

Hel2 in yeast, and ZNF598 in mammals, thus acting as a specific ribosomal collision sensor (Best et al, 2023; Chandrasekaran et al, 2019; Han et al, 2020; Ikeuchi et al, 2019; Juszkiewicz et al, 2018; Juszkiewicz and Hegde, 2017; Matsuo et al, 2017; Matsuo and Inada, 2021; Matsuo et al, 2020; Matsuo et al, 2023; Meydan and Guydosh, 2020; Narita et al, 2022; Pochopien et al, 2021; Saito et al, 2022; Tesina et al, 2020; Wu et al, 2020).

In the triggering step of RQC, Hel2 forms a K63-linked polyubiquitin chain at lysine residues K6 or K8 of ribosome protein uS10 encoded by *RPS20* around the colliding interface. This ubiquitin architecture is crucial for recruiting the RQC-triggering (RQT) complex into colliding ribosomes (Matsuo et al, 2017; Matsuo et al, 2023). The yeast RQT complex is a trimer consisting of the RNA helicase-family protein Slh1, the ubiquitin-binding protein Cue3, and Rqt4 (Matsuo et al, 2017; Matsuo et al, 2020; Matsuo et al, 2023). Two ubiquitin-binding subunits, Cue3 and Rqt4, specifically bind to the K63-linked polyubiquitin chain on uS10 (Matsuo et al, 2023). The ATPase activity of Slh1 is responsible for the subunit dissociation of polyubiquitinated collided ribosomes both in vivo (Ikeuchi et al, 2019; Matsuo et al, 2017) and in vitro (Best et al, 2023; Juszkiewicz et al, 2020; Matsuo et al, 2020; Matsuo et al, 2023; Narita et al, 2022).

This notion is conserved in mammals; ZNF598 ubiquitinates the ribosomal proteins to initiate RQC (Garzia et al, 2017; Hashimoto et al, 2020; Juszkiewicz and Hegde, 2017; Matsuo et al, 2017; Narita et al, 2022; Sundaramoorthy et al, 2017). The human RQT (hRQT) complex comprises ASCC3, ASCC2, and TRIP4/hRQT4 (Hashimoto et al, 2020; Narita et al, 2022) and dissociates the collided ribosomes dependent on the ATPase activity of ASCC3 and the ubiquitin-binding capacity of ASCC2 (Juszkiewicz et al, 2020; Narita et al, 2022). The hRQT complex-mediated subunit dissociation requires the K63-linked polyubiquitination of uS10 (Narita et al, 2022). Monoubiquitination of uS10 is not sufficient, suggesting that ZNF598 functionally marks collided mammalian ribosomes with K63-linked polyubiquitination of uS10 for trimeric hRQT complex-mediated subunit dissociation.

In the deubiquitination process of collided ribosomes in mammals, OTUD3 and USP21 antagonize ZNF598-mediated ubiquitylation of eS10. Cells lacking OTUD3 or USP21 exhibit altered RQC activity and delayed deubiquitylation of eS10 (Garshott

[1]Division of RNA and Gene Regulation, Institute of Medical Science, The University of Tokyo, Minato-Ku 108-8639, Japan. [2]Graduate School of Pharmaceutical Sciences, The University of Tokyo, Bunkyo-Ku, Tokyo, Japan. [3]Institute for Advanced Life Sciences, Hoshi University, Shinagawa-Ku, Tokyo 142-8501, Japan. [4]Division of Protein Metabolism, Institute of Medical Science, The University of Tokyo, Minato-Ku 108-8639, Japan. [5]Protein Metabolism Project, Tokyo Metropolitan Institute of Medical Science, Setagaya-Ku, Tokyo 156-8506, Japan. ✉E-mail: yoshitaka-matsuo@g.ecc.u-tokyo.ac.jp; toshiinada@ims.u-tokyo.ac.jp

et al, 2020), suggesting a functional role for deubiquitylating enzymes (DUBs) within the RQC pathway. However, little is known about how DUBs edit the polyubiquitin architectures on the uS10 to regulate the RQC induction.

Previously, various types of ribosome ubiquitination have been observed. For example, the ubiquitination of ribosomal proteins uS10, uS3, eS7, and uL23 initiates different pathways: the RQC pathway (Matsuo et al, 2017; Matsuo et al, 2020; Matsuo et al, 2023), the 18S non-functional rRNA decay (18S NRD) (Li et al, 2022; Sugiyama et al, 2019), the no-go decay (NGD) (Ikeuchi et al, 2019), and the ribophagy (Kraft et al, 2008; Ossareh-Nazari et al, 2010; Ossareh-Nazari et al, 2014), respectively. Notably, these ubiquitinations are all regulated by Ubp3 (Kraft et al, 2008; Matsuki et al, 2020), and thus, Ubp3 seems to act as the multifunctional regulator of various types of ribosome ubiquitination. In addition, hydrogen peroxide inhibits the Ubp2, leading to the accumulation of K63-polyubiquitinated ribosomes (Silva et al, 2015). Although it has been reported that Ubp3 coordinates ubiquitination and deubiquitination activities of regulatory uS3 monoubiquitination in the RQC in yeast (Jung et al, 2017), recent studies reveal that monoubiquitination of uS3 is involved in the 18S non-functional rRNA decay (NRD) (Li et al, 2022; Sugiyama et al, 2019), but not in the RQC (Matsuo et al, 2017). Therefore, the function of Ubp3 on the RQC is still unknown.

In principle, ubiquitin modifications are regulated not only by addition and elimination but also by different types of poly-ubiquitin architectures. As is well known, K48-linked and K11-linked polyubiquitin chains serve as marks for proteasomal degradation, whereas K63-linked ones work independently of the proteasome (Dikic and Schulman, 2023; Komander and Rape, 2012). Notably, there are also ubiquitin chains that combine linked types, such as K11/K48 branched chains, which promote protea-somal degradation (Kolla et al, 2022; Ohtake, 2022). In yeast, Ubc1 (mammalian homologous UBE2K) facilitates the formation of K48/K63-branched ubiquitin chains and regulates resistance to heat shock and DNA replication stress (Pluska et al, 2021). However, the function of the K48/K63 mixed or branched ubiquitin chain conjugate to the various substrates is still largely unknown.

In RQC, the crucial role of the K63-linked polyubiquitin chain on uS10 for RQT complex-mediated subunit dissociation is well-established. However, the overexpression of the K48R ubiquitin mutant leads to the reduction of the polyubiquitin chain on uS10 (Matsuo et al, 2017), suggesting that a K48-linked polyubiquitin chain may form on uS10, with an unknown function. Since the RQT complex specifically binds to the K63-linked polyubiquitin chain but not to the K48-linked ones (Matsuo et al, 2023), this observation allows us to hypothesize that the K48-linked or K48-mixed ubiquitin chain formed on uS10 may have a negative role in the RQT complex-mediated subunit dissociation.

In this study, we set out to investigate the key role of DUBs in editing polyubiquitin architectures on the uS10 for regulation of RQC activity, utilizing several unique technologies, including the Ubi-Crest assay and the Ubi-uS10 chimera system. Here, we clarify the essential functions of two DUBs, Ubp2 and Ubp3, in the editing and recycling processes of the polyubiquitin chain on uS10. These enzymatic activities significantly contribute to enhancing RQC functionality. Ubp2 specifically recycles the K63-linked polyubiquitin chain from the free 40S subunit. Conversely, Ubp3 predominantly cleaves the K48-linked di-ubiquitin and K48/K63-

mixed linkage polyubiquitin chain from translating ribosomes. Furthermore, the ubiquitin chain containing the K48-linkage of the uS10 on the colliding ribosome functions as a negative signal for the RQT-mediated ribosome dissociation process. In summary, our findings delineate the positive functions of these two DUBs in sustaining the persistent activity of RQC induction.

## Results

### A genetic screen identified two deubiquitinating enzymes in RQC

To uncover the editing mechanism of uS10 polyubiquitination on the colliding ribosome, we sought to identify the responsible DUBs using genetic screening of *Saccharomyces cerevisiae*. We monitored uS10 through immunoblotting by attaching a triple HA-tag to its C-terminal. This modification did not affect the functionality of uS10, as we observed no growth defects (Fig. EV1A). Triple HA-tagged uS10 was expressed in the single deletion mutants of 19 DUBs and compared for its ubiquitination level by immunoblotting with the wild-type strain (Fig. EV1B). These results showed that the levels of the ubiquitinated uS10 were significantly increased in the *ubp2Δ* and *ubp3Δ* mutants (Fig. EV1B). Notably, in the *ubp2Δ* mutant, the level of high-molecular-weight of ubiquitinated uS10 increased while its monoubiquitination decreased, suggesting that Ubp2 could target the polyubiquitination of uS10 (Fig. EV1B,C). All upper bands corresponding to ubiquitinated uS10 disappeared in the uS10-K6/8 R mutant, even in the absence of Ubp2 or Ubp3, indicating that these bands are formed through ubiquitination at Lys6 and/or Lys8 (Fig. EV1C). Furthermore, the double deletion of these DUBs additionally increased the ubiquitination of uS10 (Fig. 1A), indicating that these two DUBs, Ubp2 and Ubp3, play a role in editing the polyubiquitin chain on uS10.

Next, to evaluate the effects of these deubiquitinating processes on the RQC activity, we performed a well-established reporter assay containing the RQC-inducible R12 staller sequence between the *GFP* and *HIS3* open reading frames (ORF) (Dimitrova et al, 2009; Ito-Harashima et al, 2007). The R12 staller sequence induces the RQT-mediated non-canonical ribosome dissociation (Matsuo et al, 2017), resulting in the production of an incomplete nascent chain on the 60S subunit. The incomplete nascent chain is subsequently ubiquitinated by Ltn1 and sorted to the proteasome-mediated degradation pathway (Bengtson and Joazeiro, 2010; Brandman et al, 2012). Thus, the accumulation level of the incomplete nascent chain, referred to as the "arrest product", can be estimated as the RQC activity in the Ltn1 deletion background. The R12 reporter assay showed that the arrest products were gently reduced in both single deletion mutants, *ubp2Δ* and *ubp3Δ*, and synergistically reduced in the double deletion mutants, *ubp2Δubp3Δ* (Fig. 1B), in good correlation with the accumulating polyubiquitin chain of uS10 in the *ubp2Δ* and *ubp3Δ* deletion mutants.

We next evaluated the RQC activity in *ubp2Δ* and *ubp3Δ* deletion mutants using low-dose anisomycin treatment. Since anisomycin is a translation elongation inhibitor, its low-dose treatment stochastically inhibits a subset of ribosomes and allows non-inhibited ribosomes to catch up to inhibited ones, inducing the ribosome collision (Juszkiewicz et al, 2018). Thus, situations with RQC inactivity, such as cells deleted for RQC factors, display

## A

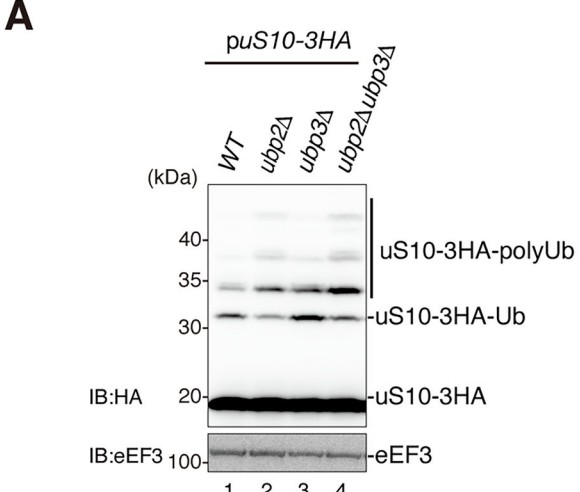

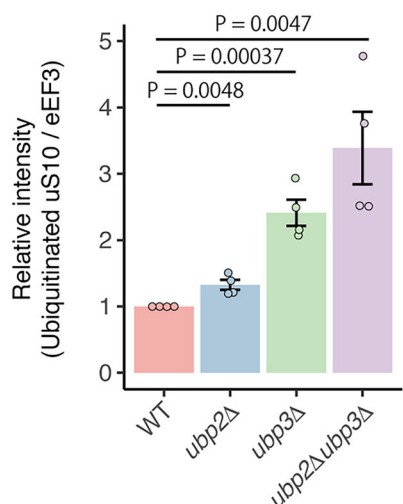

## B

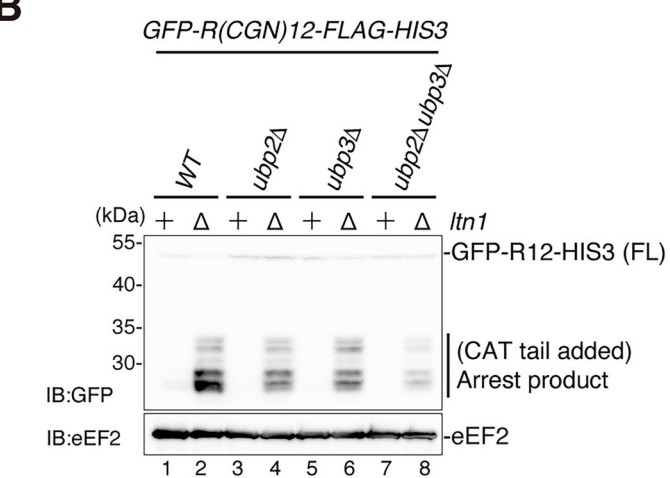

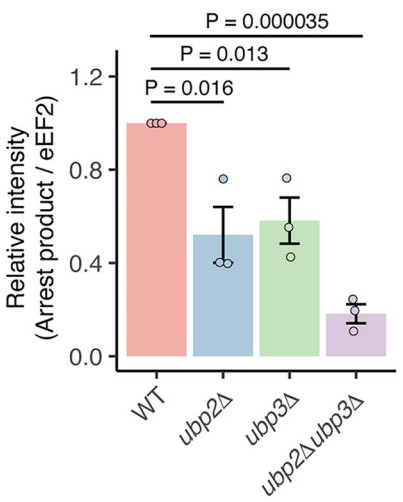

## C

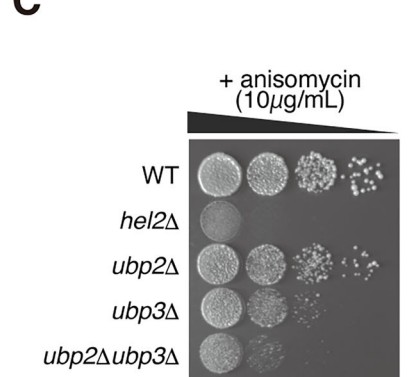

## D

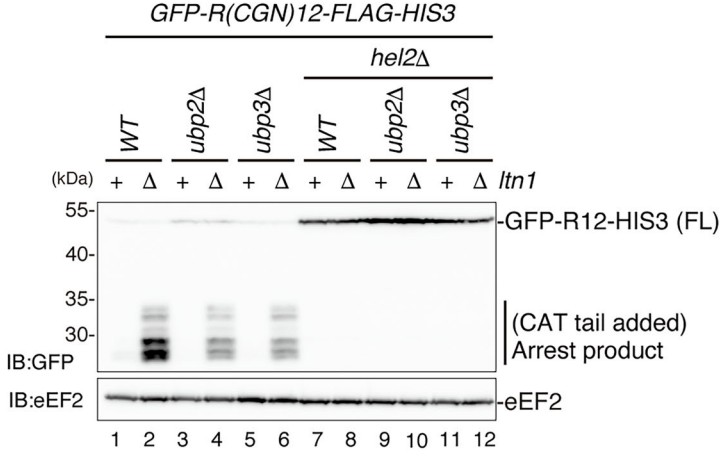

◄　**Figure 1.　Deubiquitinating enzymes promote the RQC activity.**

(**A**) The ubiquitination of uS10 increases in the *ubp2Δ*, *ubp3Δ*, and *ubp2Δubp3Δ* mutants at the steady state. The ubiquitin level of uS10-3HA derived from plasmid pST001 in the wild-type (WT), *ubp2Δ*, *ubp3Δ*, and *ubp2Δubp3Δ* mutants was detected with immunoblotting using an anti-HA antibody. The pixel intensity was measured by the plot profile tool of ImageJ. The intensity of the ubiquitinated uS10-3HA was normalized to the intensity of the eEF3 in each lane. Bar graphs represent the mean ± standard error (s.e.m.). Each dot represents an individual data point. Significance was calculated by Student's *t* tests ($n = 4$; $n$ represents the number of biological replicates). (**B**) The arrest products derived from *GFP-R(CGN)12-FLAG-HIS3* reporter reduced in deubiquitinating enzymes deletion mutant cells. The arrest products in the indicated cells: wild-type, *ltn1Δ*, *ubp2Δ*, *ubp2Δltn1Δ*, *ubp3Δ*, *ubp3Δltn1Δ*, *ubp2Δubp3Δ*, and *ubp2Δubp3Δltn1Δ*, were detected by immunoblotting using an anti-GFP antibody (top panel). The pixel intensity was measured by the plot profile tool of ImageJ. The intensity of the arrest products was normalized to the intensity of the eEF2 in each lane. Bar graphs represent the mean ± standard error (s.e.m.). Each dot represents an individual data point. Significance was calculated by Student's *t* tests ($n = 3$; $n$ represents the number of biological replicates). (**C**) Spot assay of the indicated cells in the presence of 10 μg/ml anisomycin. The indicating cells: wild-type, *hel2Δ*, *ubp2Δ*, *ubp3Δ*, *ubp2Δubp3Δ*, diluted to $OD_{600} = 0.3$, and tenfold serial dilutions were spotted and incubated at 30 °C for 2 days. (**D**) A collision sensor Hel2 is required for the arrest products in the deubiquitinating enzyme mutant cells. The arrest products in the indicated cells: wild-type, *ltn1Δ*, *ubp2Δ*, *ubp2Δltn1Δ*, *ubp3Δ*, *ubp3Δltn1Δ*, *hel2Δ*, *hel2Δltn1Δ*, *hel2Δubp2Δ*, *hel2Δubp2Δltn1Δ*, *hel2Δubp3Δ*, and *hel2Δubp3Δltn1Δ*, were detected by immunoblotting using an anti-GFP antibody (top panel). Total proteins used for immunoblottings in this Figure were prepared by TCA precipitation method. Source data are available online for this figure.

sensitivity to low-dose anisomycin treatment (Matsuo et al, 2017). The observation of growth rates in the presence of low-dose anisomycin clearly showed mild and severe growth defects in *ubp2Δ* and *ubp3Δ* single deletion mutants, respectively, and the double deletion of these DUBs displayed a synergistic effect (Fig. 1C). Ubp3 is involved in several mechanisms, including ribophagy and proteasome degradation pathway (An and Harper, 2018, 2020; Fang et al, 2016), and the *ubp3Δ* deletion mutant display a higher impact on the growth defects due to low-dose anisomycin treatment than the *ubp2Δ* deletion mutant.

Ribosome collision is sensed by the E3 ubiquitin ligase Hel2, which ubiquitinates uS10 and thereby induces the RQC pathway. Therefore, we investigated the impact of Hel2 on RQC activity and confirmed that Hel2 is required to produce arrest products in both *ubp2Δ* and *ubp3Δ* deletion mutants (Fig. 1D). Collectively, these results suggest that the two DUBs, Ubp2 and Ubp3, play positive roles in the induction of the RQC by trimming the ubiquitin chain on uS10.

## The substrate specificity of Ubp2 and Ubp3

To investigate the substrate specificity of Ubp2 and Ubp3, we separated the intracellular ribosomes by sucrose density gradient centrifugation and monitored the distribution of ubiquitination of uS10, Ubp2, and Ubp3 by western blotting. The distribution of Ubp2 and Ubp3 was different; Ubp2 was mainly detected at the top but faintly observed in the 40S fraction, whereas Ubp3 was detected in the heavy polysome fraction, which was shifted from polysome to disome fraction after MNase treatment (Fig. EV2A,B), indicating that these two DUBs act on different states of the ribosome. We next compared the ubiquitination of uS10 in wild-type, a single deletion mutant of *ubp2Δ*, *ubp3Δ* and the double deletion mutant of *ubp2Δubp3Δ* and found that the *ubp2Δ* mutant increased the polyubiquitinated uS10 in the 40S and 80S fractions (Fig. 2A–C), and *ubp3Δ* mutant increased the di-ubiquitinated uS10 in the 40S and polysome fractions (Fig. 2A–C). The double deletion mutant showed the additively increase in the polyubiquitinated uS10 (Fig. 2A–C), suggesting that Ubp2 removes the ubiquitin chain of uS10 in the free 40S and 80S ribosome, whereas Ubp3 trims the di-ubiquitin chain of uS10 in the free 40S and translating ribosome. Although the excess unassembled uS10 was observed in the uS10-3HA strain (Fig. 2A), we cannot detect ubiquitinated unassembled uS10. This may be because it is rapidly degraded by the proteasome,

or because only uS10 that is integrated into the ribosome can be ubiquitinated.

In the RQC, K63-linked polyubiquitination of uS10 stimulates the recruitment of the RQT complex; however, the accumulation of polyubiquitination of uS10 in the absence of Ubp2 and Ubp3 seemed to inhibit the RQC activity (Fig. 1). This result let us assume that different linkage types of ubiquitin chains could form on the uS10 in the *ubp2Δ* and *ubp3Δ* mutants. To confirm this possibility, we investigated the non-K63-linked polyubiquitinated uS10 in the *ubi1-4Δ* quadruple deletion strain expressing only the Ub-WT or Ub-K63R mutant (*Ub-WT* and *Ub-K63R* cells) (Finley et al, 1994; Sugiyama et al, 2019). Thus, all the linkage types of the ubiquitin chain can be observed in *Ub-WT* cells, but the K63-linked ubiquitin chains are absent in the *Ub-K63R* cells. The *ubi1-4Δ* quadruple deletion strain has only one copy of the ubiquitin gene, which causes differences in the expression level of ubiquitin compared to the wild-type strain. As a result, there are varying accumulation levels of ubiquitinated proteins. Despite this, this strain is valuable for examining the enzymatic specificity of the linkage types of ubiquitin chains. We carefully separated and monitored the ubiquitinated uS10 in *Ub-WT* and *Ub-K63R* cells by western blotting and found that non-K63-linked ubiquitin chains were formed on the uS10 (Fig. 2D, lanes 1–2); especially, the non-K63-linked ubiquitin chain was more efficiently formed in *Ub-K63R* mutant (Fig. 2D, lane2). We further examined the linkage specificity of Ubp2 and Ubp3 enzymatic activity. The deletion of *ubp3* led to the accumulation of non-K63 linkages of ubiquitin chains (Fig. 2D, lanes 5–6). In contrast, only the K63-linked ubiquitin chain was increased in the *ubp2Δ* mutant (Fig. 2D, lanes 3–4), indicating that Ubp2 trims the K63-linked polyubiquitin chain, and Ubp3 trims all types of ubiquitin chains in a K63-linkage non-specific manner.

To confirm the substrate specificity of the deubiquitinating enzymes, we conducted an in vitro deubiquitination assay. Because uS10 has two ubiquitination sites (K6 and K8 residues) (Ikeuchi et al, 2019; Matsuo et al, 2017), distinguishing between tandem ubiquitination and chained polyubiquitination using western blotting is challenging. To address this issue, we created the *uS10-K6R* and *-K8R* mutants, containing a single ubiquitination site (K8 or K6, respectively), to eliminate the formation of tandem ubiquitination; both mutants indeed retained the ubiquitination (Fig. EV2C). Although the ubiquitination of 80S ribosomes was elevated in the *ubp2Δ* cells (Fig. 2A), the 80S fractions consist of

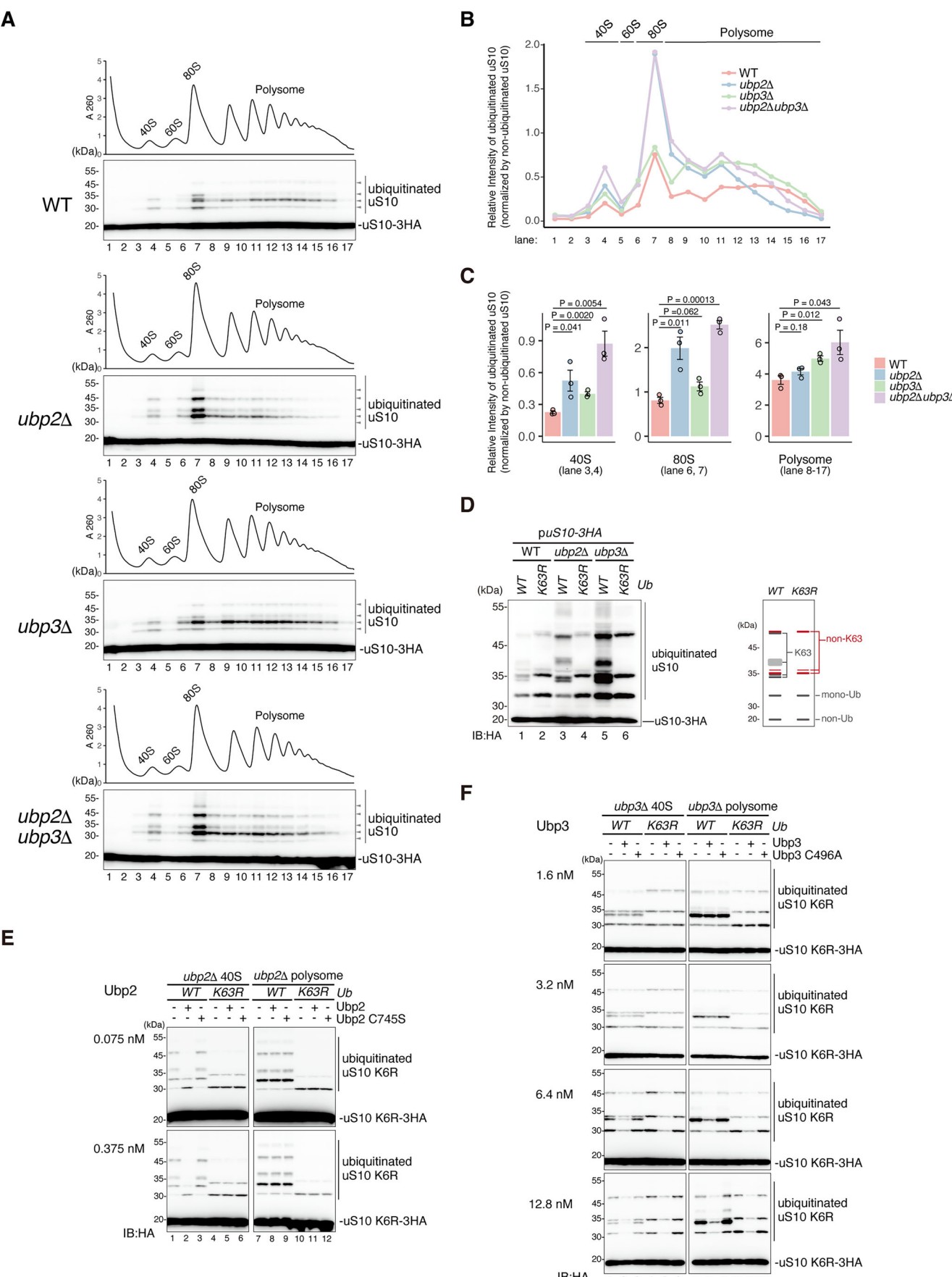

**Figure 2. Substrate specificity of the deubiquitinating enzymes Ubp2 and Ubp3.**

(A) The levels of the ubiquitinated uS10 in the deubiquitinating enzyme mutant cells. The total lysates prepared from wild-type, *ubp2Δ, ubp3Δ, ubp2Δubp3Δ* expressing uS10-3HA from plasmid pST001, and subjected to polysome analysis. The levels of the ubiquitinated uS10-3HA in the fractions were detected by immunoblotting using an anti-HA antibody. (B) Quantification of ubiquitinated uS10 in (A). The pixel intensity was measured by the plot profile tool of ImageJ. The intensity of the ubiquitinated uS10-3HA was normalized to the intensity of the non-ubiquitinated uS10-3HA in each lane. (C) Statistical quantification analysis of ubiquitinated uS10 in the different ribosome fractions. The pixel intensity was measured by the plot profile tool of ImageJ. The intensity of the ubiquitinated uS10-3HA was normalized to the intensity of the non-ubiquitinated uS10-3HA in each lane. Bar graphs represent the mean ± standard error (s.e.m.). Each dot represents an individual data point. Significance was calculated by Student's *t* tests ($n = 3$; *n* represents the number of biological replicates). (D) The non-K63-linked polyubiquitinated uS10 significantly increased in the *ubp3Δ* mutant cells but not in the *ubp2Δ* mutant cells. The ubiquitinated uS10-3HA in the *ubi1Δubi2Δubi3Δubi4Δ* mutant (referred to as *ubi1-4Δ*), as well as in the *ubi1-4Δubp2Δ* and *ubi1-4Δubp3Δ* mutant cells, was analyzed. These cells expressed either wild-type ubiquitin (Ub-WT) or the Ub-K63R mutant ubiquitin from plasmids pUB100 and pUB100-K63R, respectively. Detection was performed by immunoblotting using an anti-HA antibody. (E) In vitro deubiquitinating assay using 40S or polysome fractions of *Ub-WT ubp2Δ* strain or *Ub-K63R ubp2Δ* strains. The 40S or polysome fractions were prepared via sucrose gradient centrifugation. In the *ubi1-4Δubp2Δ* mutant cells, Ub-WT or Ub-K63R was expressed from plasmids pUB100 or pUB100-K63R, respectively, while uS10-3HA was expressed from plasmid pST001. The purified Ubp2 or Ubp2C745S (catalytic mutant) in the indicated concentration was incubated with the fractions. After the incubation at 30 °C for 60 min, the uS10-3HA was detected by immunoblotting using an anti-HA antibody. (F) In vitro deubiquitinating assay using 40S or polysome fractions of *Ub-WT ubp3Δ* strain or *Ub-K63R ubp3Δ*. The 40S or polysome fractions were prepared by sucrose gradient centrifugation. In the *ubi1-4Δubp3Δ* mutant cells, Ub-WT or Ub-K63R was expressed from plasmids pUB100 or pUB100-K63R, respectively, whereas uS10-3HA was expressed from plasmid pST001. The purified Ubp3 or Ubp3C496A (catalytic mutant) in the indicated concentration was incubated with the fractions. After the incubation at 30 °C for 60 min, the polyubiquitinated uS10-3HA was detected by immunoblotting using an anti-HA antibody. Source data are available online for this figure.

translating ribosomes and vacant ribosomes that do not contain mRNA (Heyer and Moore, 2016; Noll et al, 1973), making it difficult to discriminate the translating and vacant ribosomes in the 80S fraction. The ubiquitination of uS10, essential for the induction of RQC, occurs due to ribosomal collisions (Ikeuchi et al, 2019; Matsuo et al, 2017; Matsuo et al, 2020). Thus, the ubiquitinated ribosome is subsequently disassembled by the RQT complex (Matsuo et al, 2020), resulting in the generation of a free 40S subunit with ubiquitination. Since the translating 80S subunit is not subject to ubiquitination, the ubiquitination of the 80S ribosome observed in Fig. 2A was likely to be that of the vacant 80S subunit, where the free 40S subunit with ubiquitin chain was re-associated with the 60S subunit. Therefore, we excluded the 80S fraction for the analysis. We prepared the 40S and polysome fractions from *ubp2Δ* or *ubp3Δ* cells in the *Ub-WT* and *Ub-K63R* backgrounds expressing uS10-K6R. Subsequently, these fractions were incubated with the purified Ubp2 and Ubp3 proteins (Fig. EV2D) for the deubiquitinating reaction (Fig. 2E,F).

Ubp2-WT effectively cleaved the polyubiquitin chain on uS10 in the 40S fraction in the *Ub-WT* cells (Fig. 2E, lanes 1–2) but showed no activity in the *Ub-K63R* cells (Fig. 2E, lanes 4–5). As a control, the catalytic-dead mutant protein Ubp2-C745S (Li et al, 2020) exhibited no cleavage of the polyubiquitin chain on uS10 (Fig. 2E, lanes 3 and 6). Notably, the Ubp2-dependent cleavage of uS10 was not observed in the polysome fractions across all the combinations of genomic backgrounds (Fig. 2E, lanes 7–12). These findings strongly suggest that Ubp2 trims the K63-linked polyubiquitin chain on uS10 of the free 40S subunit but not in the translating ribosome.

Although Ubp3 exhibits lower DUB activity than Ubp2 and requires approximately a tenfold higher concentration to remove ubiquitin chains in vitro, it nonetheless cleaved polyubiquitin chains on uS10-K6R across all combinations of substrate types and genomic backgrounds (Fig. 2F). As a control, the catalytic-dead mutant protein Ubp3-C496A (Takehara et al, 2021) showed no cleavage of the polyubiquitin chain on uS10-K6R (Fig. 2F). Considering that Ubp3 was associated with heavy polysomes in vivo (Fig. EV2), we conclude that Ubp3 cleaves all types of polyubiquitin chain on uS10 in the translating ribosome.

In summary, we propose that Ubp2 trims the K63-linked ubiquitin chain on uS10 of the free 40S subunit, and Ubp3 cleaves the polyubiquitin on uS10 of translating ribosomes regardless of K63-linkage.

## K63- and K48-linkage in ubiquitin chains on uS10 are predominantly targeted by Ubp2 and Ubp3, respectively

Given the clear presence of non-K63-linked ubiquitin chains in the *Ub-K63R* mutant (Fig. 2D), our objective was to determine the linkage types of polyubiquitinated uS10 through mass spectrometry. To prepare the mass spectrometry samples without unassembled ribosome protein contamination, we extracted the ubiquitinated uS10 from the purified ribosomes using a two-step purification process (Fig. 3A). We first purified the ribosomes via Flag-tagged uL23 using an anti-Flag antibody (Fig. EV3A, lanes 3 and 9). Subsequently, free His-tagged uS10 proteins were purified using Ni-NTA resin after denaturation of the ribosome with guanidine hydrochloride (Fig. EV3A, lanes 5 and 11). To maximize the yield of ubiquitinated uS10, Hel2 was overexpressed (Fig. EV3A, lanes 4, 6, 10, 12). Finally, three identical biological replicates of ubiquitinated uS10 from wild-type, *ubp2Δ*, and *ubp3Δ* strains were prepared for mass spectrometry analysis (Fig. EV3B).

The mass spectrometric linkage analysis revealed a significant increase in K63-linked ubiquitin chains in the *ubp2Δ* mutant, along with a slight rise in K29- and K48-linked ubiquitin chains (Figs. 3C and EV3B). While these linkage types, K29-, K48-, and K63-linked, increased in the *ubp3Δ* mutant (Fig. 3B), the *ubp3Δ* mutant exhibited a notably higher proportion of K48-linked ubiquitin chains (Fig. 3C). These results suggest that Ubp2 primarily recognizes K63-linked ubiquitin chains, while Ubp3 preferentially trims K48-linked ones.

Our analysis of uS10 ubiquitin chains, using K48- and K63-linkage-specific antibodies, detected a long polyubiquitin chain in the *ubp2Δ* mutant (Fig. 3D, lanes 5 and 8). In addition, di-ubiquitinated K48- and K63-linked uS10 were observed in wild-type and the *ubp2Δ, ubp3Δ* mutant (Fig. 3D, lanes 4–9). These imply that K48/K63-mixed-linkage ubiquitin chains could be

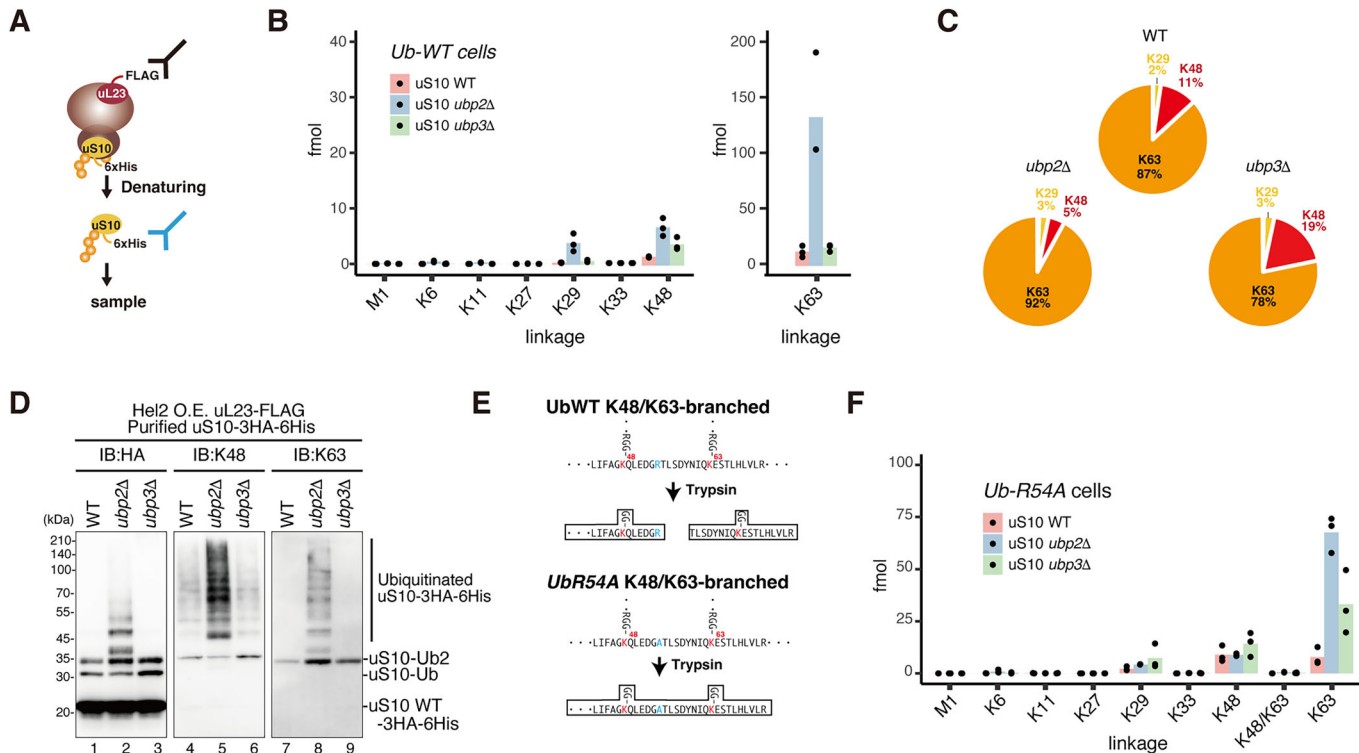

**Figure 3.  Both K63- and K48-linked polyubiquitin chains on uS10 are increased in the ubp2 and ubp3 mutants.**

(A) Schematic drawing of the uS10-3HA-His$_6$ purification. The uS10-3HA-His$_6$ was purified by a two-step affinity purification method. (B) Absolute quantification analysis of Ub chains of uS10. The uS10-3HA-His$_6$, uL23-FLAG, and Hel2-V5 were expressed from plasmids pST051, pKI191, and pST069, respectively, in the following cells: *uS10Δ*, *uS10Δubp2Δ*, or *uS10Δubp3Δ*. The uS10-3HA-His$_6$ were purified by two-step purification and analyzed by a mass spectrometer. The abundance of Ub linkages was quantified using PRM. The data show the peptide abundance (fmol) of each Ub linkage signature peptide calculated by spiking in 25 fmol heavy isotope-labeled AQUA peptides (n = 3; n represents the number of biological replicates). (C) The ratio of the linkage types in (B). (D) Both K48- and K63-linked polyubiquitin chain was increased in the *ubp2Δ* or *ubp3Δ* mutant cells. The uS10-3HA-His$_6$, uL23-FLAG, and Hel2-V5 were expressed from plasmids pST051, pKI191, and pST069, respectively, in the following cells: *uS10Δ*, *uS10Δubp2Δ*, or *uS10Δubp3Δ*. The uS10-3HA-His$_6$ were purified by two-step purification and subjected to immunoblotting using an anti-HA, K48-linkage-specific, and K63-linkage-specific anti-ubiquitin antibodies. (E) Schematic drawing of trypsin digestion for ubiquitin. Boxed peptide fragments were detected by Mass analysis. (F) Absolute quantification analysis of Ub chains of uS10 purified from the *Ub-R54A* mutants. The ubiquitinated uS10-3HA purified via a two-step purification method in the *ubi1-4Δ, ubi1-4Δubp2Δ* and *ubi1-4Δubp3Δ* mutant cells, was analyzed by mass spectrometer. These cells expressed uS10-3HA-His$_6$, uL23-FLAG, Hel2-V5, and Ub-R54A mutant ubiquitin from plasmids pST081, pST097, pST109, and pUB100-R54A. The abundance of Ub linkages was quantified using PRM. The data show the peptide abundance (fmol) of each Ub linkage signature peptide calculated by spiking in 25 fmol heavy isotope-labeled AQUA peptides (n = 3; n represents the number of biological replicates). Source data are available online for this figure.

formed on uS10, although their frequency is lower than that of the K63-linked di-ubiquitin chain.

In the subsequent investigation, we explored the potential formation of K48/K63-branched ubiquitin linkages on uS10. Typically, specific ubiquitin linkages can be identified by mass spectrometry as trypsin-digested peptides containing the GlyGly-modified lysine at a specific residue (Ohtake et al, 2019). However, the ubiquitin moiety is cleaved at R54 even in the presence of branched chains at K48 and K63 (Fig. 3E), making it challenging to distinguish the mixed or branched chains (Fig. EV3C). To overcome this issue, we employed the Ub-R54A mutant (Ohtake et al, 2016), which prevents cleavage at the R54 residue and generates the peptides containing GlyGly-modification at both K48 and K63 residues (Fig. 3E).

In our attempt to detect the formation of K48/K63-branched ubiquitin, we replicated the mass spectrometry linkage analysis using the Ub-R54A strain. However, the characteristics of the polyubiquitinated uS10 were nearly identical to those of *Ub-WT*

cells (Figs. EV3D and 3F). Notably, the branched K48/K63 linkage was not detected in this experiment (Fig. 3F), indicating that the K48/K63-linked mixed ubiquitin chain was formed on the uS10 but not K48/K63-branched one.

## K48/K63 mixed-linkage ubiquitin chain on uS10 is a substrate for the deubiquitination

In our pursuit of a deeper understanding of the polyubiquitin architecture on uS10, we conducted a UbiCRest assay employing the linkage-specific DUBs, namely AMSH* (K63-linked ubiquitin chain specific enzyme), and OTUB1* (K48-linked ubiquitin chain specific enzyme) (Hospenthal et al, 2015; Michel et al, 2015). We validated the substrate specificity of these purified enzymes, OTUB1* and AMSH*, for the K48- or K63-linked tetraubiquitin chains, respectively (Fig. EV4A). The Ubi-CRest assay conducted with the purified uS10 from the *ubp2Δ* mutant revealed the predominant formation of K63-linked ubiquitin chains on uS10

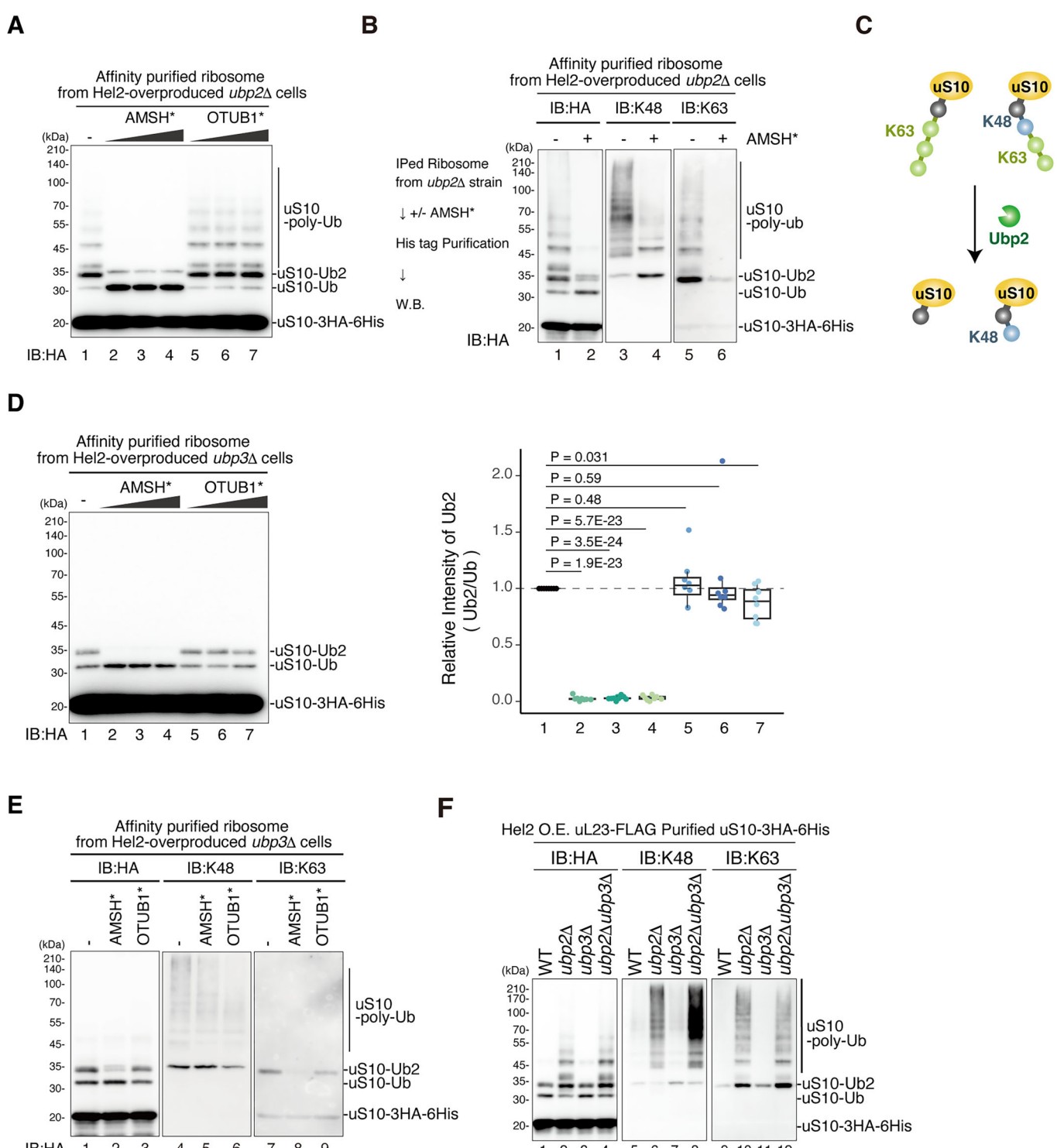

(Fig. 4A). AMSH* efficiently trims the K63-linked ubiquitin chain on uS10, leaving behind the di-ubiquitin chain after the trimming of the long chains, suggesting the formation of K48-linked di-ubiquitin (Fig. 4A, lanes 1–4). In contrast, OTUB1* cannot efficiently remove the polyubiquitin chain from uS10 (Fig. 4A, lanes 5–7).

Significantly, western blotting with the linkage-specific antibodies revealed a decrease in K48-linked polyubiquitin chains but an increase in K48-linked di-ubiquitin chains after the reaction with AMSH* (Fig. 4B, lanes 3–4). Similarly, the K63-linked di-ubiquitin chain on uS10 decreased after reaction with AMSH* (Fig. 4B, lanes 5–6). In addition, western blotting with the linkage-

◄  **Figure 4.  uS10 is ubiquitinated via K48- or K63-linked chains proximally, and via K63-linked chains distally.**

(A) The affinity-purified ribosomes containing uS10-3HA-His$_6$ from the $ubp2\Delta$ $uS10\Delta$ cells were incubated with AMSH* (K63-linkage-specific deubiquitinase) and OTUB1* (K48-linkage-specific deubiquitinase). Immunoblotting was performed using an anti-HA antibody. (B) The affinity-purified ribosomes with uS10-3HA-His$_6$ from the $ubp2\Delta uS10\Delta$ mutant cells expressing uS10-3HA-His$_6$, uL23-FLAG, and Hel2-V5 from plasmids pST051, pKI191, and pST069, respectively, cells were incubated with AMSH*. Immunoblotting was performed using an anti-HA antibody (left), K48-linkage-specific anti-ubiquitin antibody (middle), and K63-linkage-specific anti-ubiquitin antibody (right). (C) Schematic drawing of ubiquitin chain linkage on uS10 and how Ubp2 trim ubiquitin chains. (D) The affinity-purified ribosomes containing uS10-3HA-His$_6$ from the $ubp3\Delta uS10\Delta$ mutant cells expressing uS10-3HA-His$_6$, uL23-FLAG, and Hel2-V5 from plasmids pST051, pKI191, and pST069, respectively, were incubated with AMSH* and OTUB1*. Immunoblotting was performed using an anti-HA antibody (Left panel). The ratio between mono- and di-ubiquitin of uS10 were plotted (Right panel). The pixel intensity was measured by the plot profile tool of ImageJ. Boxplots represent the distribution of values for each lane. The central line in each box indicates the median (50th percentile). The lower and upper bounds of the box correspond to the first (25th percentile) and third quartiles (75th percentile), respectively. The whiskers extend to the minimum and maximum values within 1.5 times the interquartile range (IQR) from the lower and upper quartiles. Individual data points, including potential outliers beyond this range, are overlaid as open circles with jitter for visualization. The intensity of the di-ubiquitinated uS10-3HA was normalized to the intensity of the mono-ubiquitinated uS10-3HA in each lane. Significance was calculated by Student's $t$ tests ($n = 8$; $n$ represents the number of technical replicates). (E) The affinity-purified ribosomes containing uS10-3HA-His$_6$ from $ubp3\Delta uS10\Delta$ mutant cells expressing uS10-3HA-His$_6$, uL23-FLAG, and Hel2-V5 from plasmids pST051, pKI191, and pST069, respectively, were incubated with AMSH* and OTUB1*. Immunoblotting of uS10-3HA-His$_6$ performed using an anti-HA antibody (left), K48-linkage-specific anti-ubiquitin antibody (middle), and K63-linkage-specific anti-ubiquitin antibody (right). (F) Both K48- and K63-linked polyubiquitin chain was increased in the $ubp2\Delta$, $ubp3\Delta$, and $ubp2\Delta ubp3\Delta$ mutant cells. The uS10-3HA-His$_6$ purified via two-step purification from the following mutant cells: $uS10\Delta$, $uS10\Delta ubp2\Delta$, $uS10\Delta ubp3\Delta$, or $uS10\Delta ubp2\Delta ubp3\Delta$, expressing uS10-3HA-His$_6$, uL23-FLAG, and Hel2-V5 from plasmids pST051, pKI191, and pST069, respectively, were subjected to immunoblotting using an anti-HA, K48-linkage-specific, and K63-linkage-specific anti-ubiquitin antibodies. Source data are available online for this figure.

specific antibodies confirmed the presence of K48- and K63-linked di-ubiquitin on the purified uS10 (Fig. EV4B). Collectively, two types of polyubiquitin architectures: K63-homo-linkage and K48/K63-mixed-linkage, accumulate in the $ubp2\Delta$ mutant. In the K48/K63 mixed ubiquitin chain (Fig. 4B), K48-linkage predominantly appears at the proximal site (Fig. 4C), with a possibility of distal ubiquitin chains also forming the K48 linkage.

The Ubi-CRest assay conducted with the purified uS10 from the $ubp3\Delta$ mutant revealed that the di-ubiquitin chain on uS10 is predominantly K63-linked but contains a small amount of K48-linked form (Fig. 4D, lanes 2–4). The di-ubiquitin chain on uS10 was almost eradicated by the AMSH*. By contrast, the di-ubiquitin chain slightly decreased along with the increase of monoubiquitin by OTUB1* (Fig. 4D, left panel: lanes 5–7). The quantification of the ratio between di-Ubi and mono-Ubi showed a slight reduction of di-ubiquitination of uS10 after reaction with OTUB1* (Fig. 4D, right panel). Following the reaction with AMSH*, the K48-linked di-ubiquitin chain did not increase (Fig. 4E, lanes 4–5), and the K63-linked di-ubiquitin chain on uS10 decreased (Fig. 4E, lanes 7 and 8). Post-reaction with OTUB1*, the K48-linked di-ubiquitin chain decreased (Fig. 4E, lanes 4 and 6), and the K63-linked di-ubiquitin chain on uS10 did not increase (Fig. 4E, lanes 7 and 9).

To further scrutinize the substrate specificity of Ubp2 and Ubp3, we investigated the linkage of uS10 in the $ubp2\Delta ubp3\Delta$ double deletion mutant (Fig. 4F). The level of K48-linked di- and polyubiquitin were higher in the $ubp2\Delta ubp3\Delta$ mutant than in the $ubp2\Delta$ mutant (Fig. 4F, lanes 6–8). Conversely, the levels of K63-linked di- and poly-ubiquitin in $ubp2\Delta$ and $ubp2\Delta ubp3\Delta$ mutants were almost identical (Fig. 4F, lanes 10–12). Given that the K48/K63-mixed-linkage ubiquitin chains on uS10, via K48-linked di-ubiquitin, were increased in the $ubp2\Delta ubp3\Delta$ mutant cells, these results support the model that Ubp2 trims the K63-linked chain, and Ubp3 trims K48-linkage ubiquitin chains. Collectively, these results suggest the linkage of the polyubiquitin chain on uS10 is primarily K63-linkage, but K48/K63-mixed-linkage ubiquitin chains are formed via K48-linked di-ubiquitin chain (uS10-Ub-K48-K63$_N$: Fig. EV4C) at a low frequency.

Considering that Ubp2 specifically targets the free 40S subunit (Fig. 2E), and Ubp3 is mainly associated with polysomes

(Fig. EV2B), we propose the model in which Ubp2 trims the K63-linked ubiquitin chain for recycling, and Ubp3 removes K48/K63-mixed-linkage ubiquitin chain at the translating ribosomes.

## K29-, K48, and K63-linked ubiquitin chains formed on the ubiquitin-fused uS10 (Ub-uS10)

The architecture of the K48/K63-mixed-linkage ubiquitin chain of uS10 suggests that the K48-linked di-ubiquitin chain forms at the proximal site, and the K63-linked ubiquitin is further elongated from this K48-linked di-ubiquitin chain. To reconstitute this polyubiquitin architecture, we aimed to establish the Ub-uS10 chimera system, which enables us to manipulate the first linkage type at the proximal site on uS10. The ubiquitin with G76V mutation was fused to residues 8–121 of uS10, lacking two lysine residues for ubiquitination (Fig. 5A, top), behaving as the mono-ubiquitinated uS10 analog. Thus, in this system, we can manipulate the first linkage type using Ub-uS10 mutants (Fig. 5A).

The fusion of ubiquitin to uS10 did not affect the functionality of uS10, as we observed no growth defects (Fig. EV1A). To further confirm the functionality of the Ub-uS10 chimera protein in RQC, we conducted the R12 reporter assay. In the absence of Ltn1, arrest products were observed, indicating that Ub-uS10 retains functionality for RQC (Fig. 5B, lane 1).

To investigate the effects of the three linkage types (K29-, K48-, and K63-linkage) of uS10 on RQC activity, we performed lysine-to-arginine substitutions with all combinations (Fig. 5A). In the R12 reporter assay, arrest products were not detected in all mutants with the K63R combinations (Ub-K63R, Ub-K29/K63R Ub-K48/K63R, Ub-K29R/K48/K63R) (Fig. 5B, lanes 4, 6–8), whereas K29R and K48R mutants of Ub-uS10 still produced arrest products (Fig. 5B, lanes 2, 3, 5). Consistent with this result, the K63R mutation in any combination with K29R or K48R conferred sensitivity to anisomycin (Fig. 5C). These results indicate that the K63-linked polyubiquitination of Ub-uS10 is essential and sufficient for RQC induction.

We conducted a detailed analysis of the ubiquitination process of Ub-uS10 with a specific linkage. Ub-uS10 with K63-linked monoubiquitin was detected in Ub-uS10 wild-type but in none of

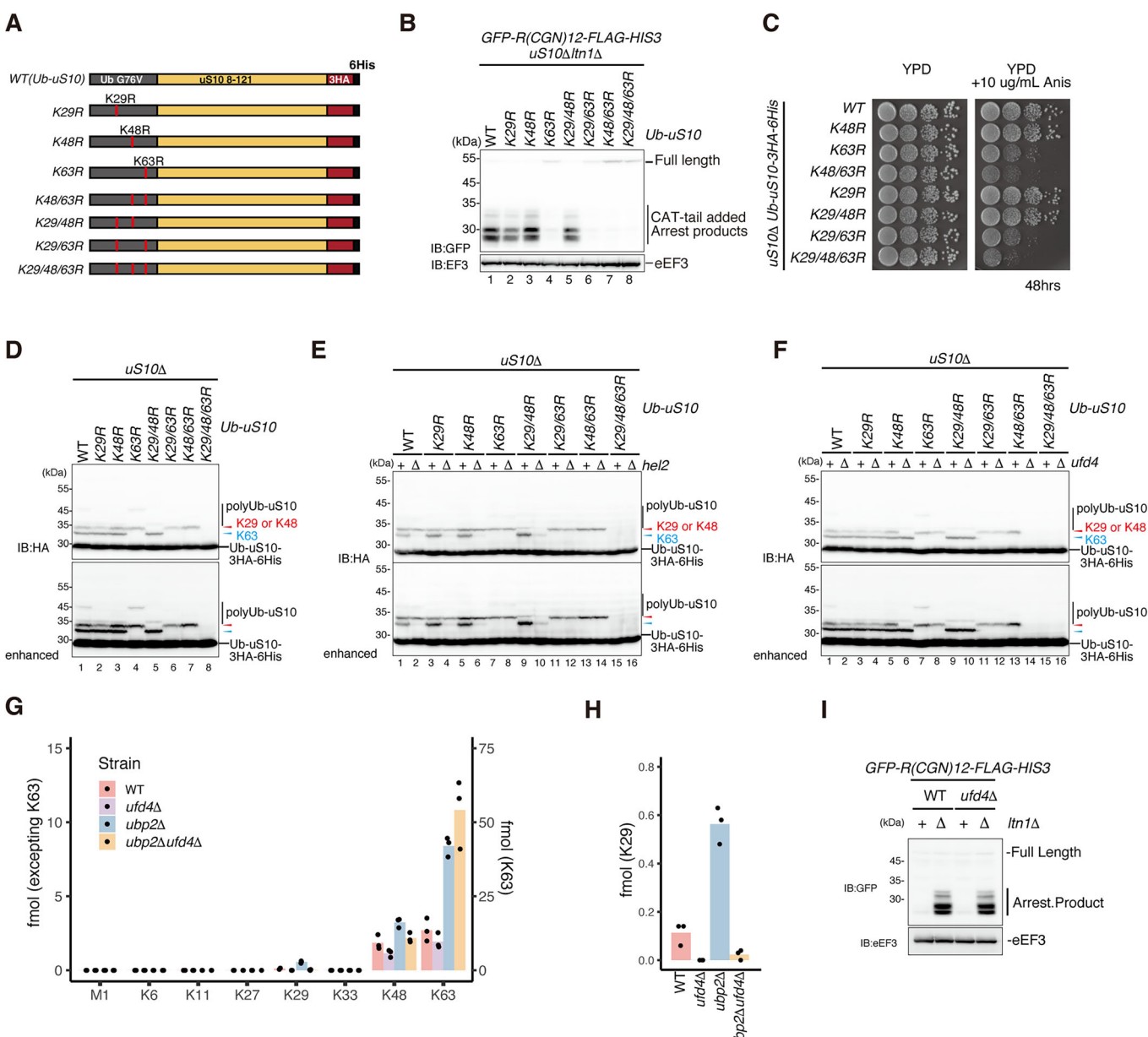

**Figure 5. Investigation of the effects of proximal linkage type on the RQC activity using ubiquitin-fused uS10 (Ub-uS10) system.**

(A) Schematic drawing of the Ub-uS10-3HA-His$_6$ construct. To construct Ub-uS10, the ubiquitin with G76V mutation was fused to 8–121 residues of uS10 that lack two lysine residues for Hel2-mediated ubiquitination. Two tag sequences, HA and six histidine residues (His$_6$) were inserted into C-terminus. (B) RQC is intact in cells expressing Ub-uS10 constructs. The arrest products derived from *GFP-R(CGN)12-FLAG-HIS3* reporter in the *uS10Δltn1Δ* cells expressing Ub-uS10 constructs (A) from plasmids pST158, pST159, pST160, pST161, pST162, pST163, pST164, or pST165 were detected by immunoblotting using an anti-GFP antibody (top panel). (C) Spot assay of *uS10Δ* cells expressing Ub-uS10 constructs (A) from plasmids pST158, pST159, pST160, pST161, pST162, pST163, pST164, or pST165 in the absence or presence of 10 μg/ml anisomycin. Cells diluted to OD$_{600}$ = 0.3 and tenfold serial dilutions were spotted and incubated at 30°C for two days. (D) The analysis of the ubiquitination of Ub-uS10 constructs (A) expressing from plasmids pST158, pST159, pST160, pST161, pST162, pST163, pST164, or pST165 in the *uS10Δ* cell. Protein samples were prepared from the indicated cell expressing Ub-uS10 wild-type or the mutations. The polyubiquitinated uS10 was detected with immunoblotting using an anti-HA antibody. (E) Hel2 is responsible for the K63-linked ubiquitination of Ub-uS10. The analysis of the ubiquitination of Ub-uS10 constructs (A) expressing from plasmids pST158, pST159, pST160, pST161, pST162, pST163, pST164, or pST165 in the *uS10Δ* or *uS10Δhel2Δ* cell. The ubiquitinated uS10 was detected as in (D). (F–H) Ufd4 is solely responsible for the formation of the K29-linked polyubiquitin chain. (F) The analysis of the ubiquitination of Ub-uS10 constructs (A) expressing from plasmids pST158, pST159, pST160, pST161, pST162, pST163, pST164, or pST165 in the *uS10Δ* or *uS10Δufd4Δ* cell. The ubiquitinated uS10 was detected as in (D). (G, H) Absolute quantification analysis of Ub chains of uS10 purified from the indicated mutants. The abundance of Ub linkages was quantified using PRM. The data show the peptide abundance (fmol) of each Ub linkage signature peptide calculated by spiking in 25 fmol heavy isotope-labeled AQUA peptides (*n* = 3; *n* represents the number of biological replicates). (I) Ufd4 does not serve in RQC. The arrest products derived from *GFP-R(CGN)12-FLAG-HIS3* reporter in the *ufd4Δ* or *ufd4Δltn1Δ* cells were detected by immunoblotting using an anti-GFP antibody (top panel). Total proteins used for all immunoblotting in this Figure were prepared by TCA precipitation method (B, D, E, F, I). Source data are available online for this figure.

the mutants containing Ub-K63R mutation (Fig. 5D, lanes 1, 4, 6–8). The bands corresponding to Ub-uS10 with K29-linked ubiquitin were observed in Ub-K48R/K63R-uS10 mutant (Fig. 5D, lane 7), and the bands corresponding to Ub-uS10 with K48-linked ubiquitin were found in Ub-K29R/K63R-uS10 mutant (Fig. 5D, lane 6). Both sets of bands were identical in size and were eliminated in Ub-K29R/K48R/K63R-uS10 triple mutant (Fig. 5D, lane 8). Di-ubiquitinated Ub-uS10 with the K29 or K48-linkage was also detected in the *Ub-K63R* mutant (Fig. 5D, lane 4).

We next examined the roles of E3 ligases involved in the ubiquitination of Ub-uS10. The level of Ub-uS10 with the K63-linked mono- and di-ubiquitin was strongly reduced by the *HEL2* deletion (Fig. 5E, lanes 9–10), confirming an essential role of Hel2 in forming K63-linked ubiquitin on Ub-uS10. The levels of Ub-uS10 with the K48-linked monoubiquitin were not affected in the *hel2Δ* background (Fig. 5E, lanes 11 and 12), indicating that Hel2 exclusively forms K63-linked ubiquitin on Ub-uS10. Ufd4 is an E3 ligase responsible for K29-linked ubiquitination (Tsuchiya et al, 2013). The Ub-uS10 with K29-linked monoubiquitin disappeared in the *ufd4Δ* mutant background (Fig. 5F, lanes 13 and 14), confirming that Ufd4 forms K29-linked ubiquitin on Ub-uS10. Mass spec analysis revealed that K29-linked polyubiquitin chains increased in the *ubp2Δ* but were abolished in the *ufd4Δ* and *ufd4Δubp2Δ* mutants (Figs. EV5 and 5G,H). These results further confirm that Ufd4 forms, and Ubp2 trims, K29-linked ubiquitin chain on uS10. However, the deletion of *ufd4* in the RQC reporter assay did not show any effects on the production of arrest products (Fig. 5I), indicating that the Ufd4-mediated K29-linked ubiquitin chain is not involved in the RQC induction.

## The reconstitution of K48/K63-mixed ubiquitin chain-conjugated colliding ribosomes

To focus on the function of the K48/K63-mixed-linkage ubiquitin chain in colliding ribosomes, we sought to reconstitute it in vitro using the Ub-uS10 chimera system. Colliding ribosomes were prepared by the well-established in vitro translation system using an RQC-inducible arrest sequence in the *SDD1* mRNA (Matsuo et al, 2020). Cell-free in vitro translation extracts were generated from the Ub-K48only-uS10 and the Ub-K63only-uS10 mutant strains. The Ub-K48only-uS10 and Ub-K63only-uS10 mutants retain the K48 and K63 single lysine residue in the ubiquitin region, respectively. Other lysine residues in the ubiquitin region are all substituted with arginine residues, allowing the formation of K48- or K63-linkage at the proximal site of uS10.

The model mRNA coding N-terminal His-tagged Sdd1 protein was translated in vitro using the extract prepared from the Ub-K63only-uS10 mutant strain. After the reaction, the ribosome nascent chain complexes (RNCs) were affinity-purified using the N-terminal His-tag of the nascent chains and separated by sucrose density gradient centrifugation. This result clearly demonstrated the efficient ubiquitination of colliding ribosomes (Fig. 6A). To further investigate the architecture of these ubiquitin chains on the RNCs, we conducted the UbiCRest assay using AMSH*. All ubiquitinated bands of Ub-uS10 from di- to longer polyubiquitination were diminished by the treatment with AMSH* (Fig. 6B), indicating the formation of homo K63-linked polyubiquitin chains on the colliding ribosomes.

In the case of Ub-K48only-uS10, the polyubiquitination on Ub-uS10 in the colliding ribosome drastically decreased (Fig. 6C). Initial ubiquitination with K48-linkage was efficient, but the subsequent elongation of ubiquitin chains was strongly inhibited (Fig. 6C). A small amount of polyubiquitination formed on the K48-linked Ub-uS10 was confirmed as K63-linked ubiquitin chains by UbiCRest with AMSH* (Fig. 6D). These K63-linked ubiquitin chains on the K48-linked Ub-uS10 decreased in the *hel2Δ* mutant (Fig. 6E), concomitantly with an increase in initial K48-linked ubiquitin on Ub-uS10 (Fig. 6E). These results suggest that Hel2 can elongate the K63-linked ubiquitin chain on the K48-linked Ub-uS10 but cannot conjugate the initial K48-linked ubiquitin on the Ub-uS10.

As previously reported, the RQT complex recognizes K63-linked ubiquitin chains but not K48-linked ubiquitin chains, raising the question of whether K48/K63-mixed-linkage ubiquitin chains are recognized by the RQT complex. To address this query, we conducted a pull-down assay between the RQT complex and the K48/K63-mixed-linkage tetraubiquitin chains. Under previous findings, the RQT complex bound to the K63-linked tetraubiquitin chains but not to K48-linked or K48/K63-mixed-linkage tetra-ubiquitin chains (Fig. 6F). We further confirmed that the RQT complex was stably associated with colliding ribosomes in the Ub-K63only-uS10 mutant strain but not in the Ub-K48only-uS10 mutant strain (Fig. EV6).

In conclusion, the K48-linkage on the uS10 plays a negative role in RQC induction with composite effects, inhibiting the formation of K63-linked ubiquitin chains on uS10 and impeding the recruitment of the RQT complex into colliding ribosomes.

## The RQT complex dissociates the colliding ribosomes with the K63-linked but not the K48/K63-mixed polyubiquitin chain on uS10

Since we succeeded in reconstituting the colliding ribosomes conjugated with a K48/K63-mixed-linkage ubiquitin chain on uS10 by an in vitro system (Fig. 6C), we next examined whether the RQT complex can split the colliding ribosome marked with a K48/K63-mixed-linkage ubiquitin chain. The colliding ribosomes were prepared from translation extracts derived from the Ub-K48only-uS10 or the Ub-K63only-uS10 mutant strains, generating the colliding ribosome with two types of polyubiquitin architectures, K63-linked or K48/K63-mixed-linkage ubiquitin chain on uS10. These RNCs were treated with or without the RQT complex in the presence of ATP for the splitting reaction and separated by sucrose density gradient centrifugation. As expected, the colliding ribosomes conjugated with a K48/K63-mixed-linkage ubiquitin chain were not affected after the addition of the RQT complex in the presence of ATP (Fig. 7A). By contrast, the peaks of tri- and tetra-somes in the gradient profile of Ub-K63only-uS10 mutant were decreased, and the ubiquitinated uS10 in the colliding ribosome conjugated homo K63-linked ubiquitin chain shifted from the tri- and tetra-some fractions to the 40S fraction (Fig. 7B), indicating that the colliding ribosome K63-linked ubiquitin chains were dissociated by the treatment with RQT complex. These findings suggest that the K48-linked and K48/K63-mixed-linkage ubiquitin chain on uS10 strongly inhibits RQT complex-mediated ribosome dissociation, even when the ribosome is collided.

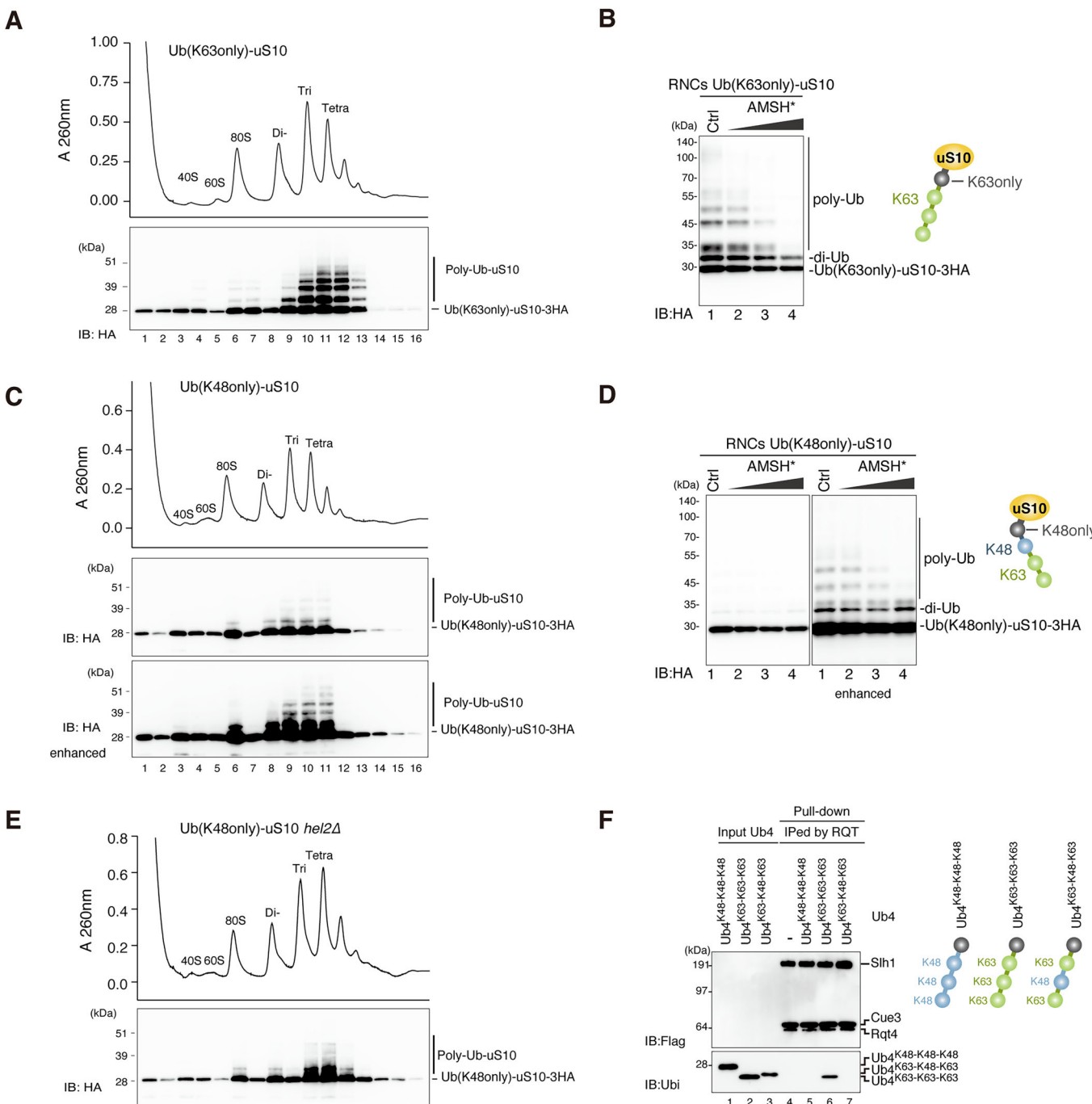

**Figure 6. The in vitro reconstitution of the K48/K63-mixed ubiquitin chain-conjugated colliding ribosomes.**

(A, C, E) In vitro reconstitution assay for colliding ribosomes: The purified RNCs from the in vitro translation (IVT) reaction of the *His-SDD1* model mRNA using the IVT extract prepared from Ub-K63only-uS10 mutant strain (A): *uS10Δski2Δ* expressing Ub-K63only-uS10-3HA from plasmid pST320, Ub-K48only-uS10 mutant strain (C): *uS10Δski2Δ* expressing Ub-K48only-uS10-3HA from plasmid pST321, or Ub-K48only-uS10 expressing *hel2Δ* mutant strain (E): *uS10Δski2Δhel2Δ* expressing Ub-K48only-3HA from plasmid pST321, were separated by sucrose density gradient centrifugation and detected by UV absorbance at a wavelength of 260 nm. HA-tagged uS10 in each fraction was detected by immunoblotting using an anti-HA antibody. (B, D) Ub-CRest assay: The purified RNCs from the in vitro translation (IVT) of *His-SDD1* model mRNA using the IVT extract prepared from Ub-K63only-uS10 mutant strain (B) or Ub-K48only-uS10 mutant strain (D), were incubated with AMSH*. Immunoblotting was performed using an anti-HA antibody. (F) Pull-down assay of the trimer RQT complex with the Ub4 (K63-K63-K63), Ub4 (K48-K48-K48), and the Ub4 (K63-K48-K63): The RQT complex was immobilized on IgG magnetic beads and mixed with each Ub4. After the binding and washing step, the proteins in the final elution were separated by 10% Nu-PAGE and detected by anti-ubiquitin and anti-FLAG antibodies. Source data are available online for this figure.

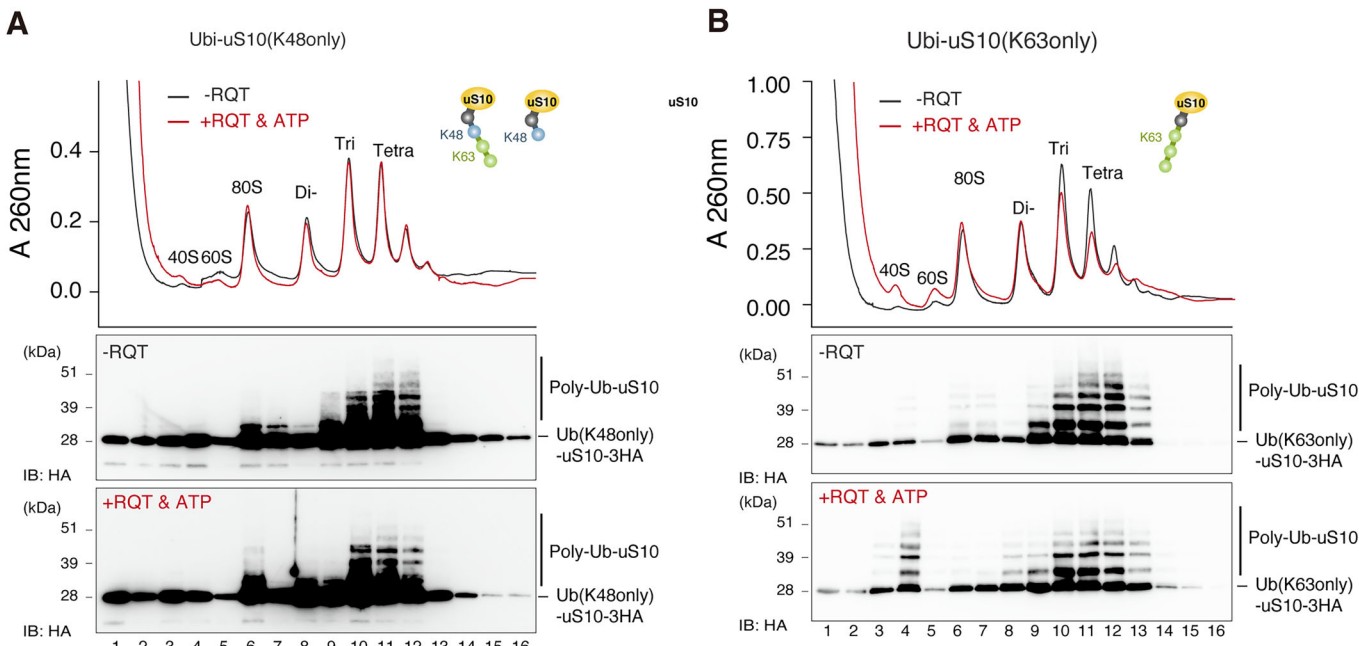

**Figure 7. RQT complex dissociates the collided ribosome with the K63-linked polyubiquitin chain but not K48-K63-mixed polyubiquitin chain on Ub-uS10.**

(A, B) In vitro splitting assay using different linkage types conjugated colliding ribosomes: The K48/K63-mixed linkage or the K63-linked polyubiquitin chain-conjugated RNCs were purified IVT reaction of *His-SDD1* model mRNA using the extract prepared from Ub-K48only-uS10 mutant strain (**A**): *uS10Δski2Δ* expressing Ub-K48only-uS10-3HA from plasmid pST321, or Ub-K63only-uS10 mutant strain (**B**) *uS10Δski2Δ* expressing Ub-K63only-uS10-3HA from plasmid pST320. Subsequently, the purified RNCs were mixed with or without the RQT complex in the presence of ATP and incubated for 45 min at 25 °C. After the reaction, RNCs were separated by sucrose density gradient centrifugation. HA-tagged uS10 in each fraction was detected by immunoblotting using an anti-HA antibody. Source data are available online for this figure.

# Discussion

The ribosome ubiquitination is a pivotal event in activating ubiquitin-mediated pathways of translational control. Various types of ribosome ubiquitination are observed; for example, the ubiquitination of ribosomal proteins uS10, uS3, eS7, and uL23 initiates different pathways: the RQC pathway (Matsuo et al, 2017; Matsuo et al, 2020; Matsuo et al, 2023), the 18S non-functional rRNA decay (18S NRD) (Li et al, 2022; Sugiyama et al, 2019), the no-go decay (NGD) (Ikeuchi et al, 2019), and the ribophagy (Ossareh-Nazari et al, 2014), respectively. However, the polyubiquitin architectures and their functions in each response have remained enigmatic. Here, we focused on the ubiquitination of uS10 in the RQC pathway and found that two DUBs, Ubp2 and Ubp3, play key roles in editing and recycling polyubiquitin chains on uS10, thereby contributing to maintaining persistent RQC activity.

Ubp2 trims specifically the K63-linked chains of uS10 on the free 40S subunit but not on the polyribosomes (Fig. 2E). This indicates that Ubp2 recycles the K63-linked polyubiquitin chain and renews the 40S subunit after ribosome dissociation. We further found the possibility that the translating or vacant ribosomes in 80S fraction may also be targeted by the Ubp2 (Fig. 2A).

Although Ubp2 is targeted to the 40S subunit after ribosome dissociation, deletion of Ubp2 leads to a reduction in RQC activity (Fig. 1B). It is well-established that ubiquitinated collided ribosomes are disassembled by the RQT complex (Matsuo et al,

2020; Matsuo et al, 2023), and cryo-EM analysis has revealed that the RQT complex remains bound to the 40S subunit after subunit dissociation (Best et al, 2023). Cue3 and Rqt4, components of the RQT complex, interact with K63-linked ubiquitin chains attached to the collided ribosomes, thereby recruiting the RNA helicase Slh1 to the stalled ribosomes (Matsuo et al, 2023). Therefore, it is plausible that the RQT complex remains tethered to the 40S subunit via ubiquitin chains even after subunit dissociation. This suggests that the Ubp2-mediated removal of K63-linked ubiquitin chains from the 40S subunit may facilitate recycling of the RQT complex. Consequently, the loss of Ubp2 could reduce the pool of available RQT complexes, thereby impairing RQC activity.

As recently reported, another DUB, Otu2, is involved in recycling the monoubiquitination of eS7 to renew the ubiquitinated 40S subunit for the next round of translation (Takehara et al, 2021). Cryo-EM analysis reveals that Otu2 specifically binds to the interface side of the 40S subunit (Ikeuchi et al, 2023), where it associates with the 60S subunit in the 80S ribosome, explaining the substrate specificity for mono-ubiquitinated eS7 only on the free 40S subunit (Ikeuchi et al, 2023). Thus, Ubp2 may also interact with the interface side of the 40S subunit, thereby discriminating against 80S ribosomes, in analogy with Otu2. However, as shown in Fig. EV2A, Ubp2 is not stably associated with the 40S subunit. Therefore, the technical development of the generation of the Ubp2-40S complex will allow us to analyze the further structural aspects and uncover the substrate specificity of Ubp2 in the future.

Ubp3 is associated with translating ribosomes (Fig. EV2B) and predominantly eliminates the K48-linked ubiquitin chains (Fig. 3B,C), contributing to the promotion of RQC activity (Fig. 1B–D). However, in vitro DUB assay revealed that Ubp3 seemed to remove all types of ubiquitin linkage on uS10, including monoubiquitin. These observations showed that Ubp3 can cleave the covalent bond between uS10 and the proximal ubiquitin, as it can also remove all types of ubiquitin linkage on the uS10. Therefore, we think that the enzymatic activity of Ubp3 is not specific to the K48-linked ubiquitin chain. In the different situations, Ubp3 can remove other linkage types of polyubiquitination, as previously reported (Fang et al, 2016; Li et al, 2020). Although K48/K63-mixed linkage poly-ubiquitin chains are formed on the colliding ribosome same as K63-linked polyubiquitin chain (Fig. 6A,C), the K48-linked di-ubiquitination can be observed at the non-colliding ribosome (80S fraction) (Fig. 6C), suggesting that initial K48-linked ubiquitination on the Ub-uS10 did not seem to depend on the ribosome collision. Ubp3 may target these K48-linked di-ubiquitinations in the non-colliding translating ribosomes. Furthermore, these ubiquitin chains containing K48-linkage act as a negative signal for the RQT-mediated ribosome dissociation process (Fig. 7A), preventing the recruitment of the RQT complex (Fig. 6F). Considering these observations, on the colliding ribosomes, K63-linked ubiquitinated ribosomes are immediately recognized by the RQT complex, leading to subunit dissociation (Matsuo et al, 2020; Matsuo et al, 2023). Therefore, Ubp3 cannot remove the K63-linked ubiquitin chain in this situation. In vitro DUB assays (Fig. 2E,F) have shown that the enzymatic activity of Ubp3 is substantially lower than that of Ubp2. Due to this weak activity, it is possible that Ubp3 cannot effectively compete with the RQT complex for binding to K63-linked chains on collided ribosomes. In contrast, K48/K63-mixed ubiquitinated ribosomes are not recognized by the RQT complex, allowing Ubp3 to access the K48/K63-mixed linkage ubiquitin chains even in the colliding ribosomes. These explain why Ubp3 appears to remove the K48-linked ubiquitin chain preferentially on the translating ribosomes.

Recent studies have revealed that the accumulation of ribosome collisions induces various cellular responses, including integrated stress response (ISR) and ribotoxic stress response (RSR). This suggests that there may be a mechanism that actively accumulates ribosome collisions to trigger the stress response. The K48/K63 ubiquitin chains on uS10 are thought to be part of this regulatory mechanism. It has been reported that under stress conditions such as ER stress, the expression level of Ubp3 decreases, leading to enhanced ribosome ubiquitination (Matsuki et al, 2020). Under such conditions, RQT-mediated ribosome dissociation might be inhibited by the K48/K63 chains on uS10, causing the accumulation of ribosome collisions. This could promote the downstream cellular responses. Further investigation is needed to confirm these hypotheses.

Unfortunately, although we cannot identify the responsible E2/E3 ligase to conjugate the K48-linked ubiquitin with uS10 on the colliding ribosome, these unknown factors could act as a positive regulator for the cellular responses against the accumulation of ribosome collision, such as the ISR and RSR (Meydan and Guydosh, 2020; Wu et al, 2020). Thus, we must identify these factors in the future. The elimination of the K48-linkage containing ubiquitin chain from uS10 on the colliding ribosome by Ubp3

could give a new chance to produce the homo K63-linked polyubiquitin chain by Hel2, stimulating the RQT complex-mediated ribosome dissociation process. Collectively, the editing process of the polyubiquitin architecture by Ubp3 determines the fate of the response to ribosome collision, which route is chosen, clearing of ribosome collision, or induction of stress responses. This hypothesis should be further investigated in future in vivo studies.

In mammals, two DUBs, OTUD3 and USP21, antagonize ZNF598-mediated 40S ubiquitination, and its regulatory function in RQC was proposed (Garshott et al, 2020). However, the roles of DUBs in RQC and the steps in RQC regulated by deubiquitination remain elusive. The K63-linked polyubiquitin chain on uS10 is also crucial for the human RQT complex-mediated splitting (Narita et al, 2022). Therefore, the substrate specificity of the deubiquitinates on the ubiquitinated uS10 or eS10 should be investigated to understand the functional role of DUBs within the RQC pathway. USP21 cleaves the di-ubiquitin chain nonspecifically (Mevissen et al, 2013), therefore, it may cleave the non-K63-linked polyubiquitin chain on uS10 as an analogous function to Ubp3 in yeast. USP10, a mammalian homolog of Ubp3, forms a G3BP1-Family-USP10 deubiquitinase complex and rescues ubiquitinated 40S subunits of stalled ribosomes from lysosomal degradation (Meyer et al, 2020). It is also important to investigate whether USP10 could trim the polyubiquitin chain on uS10 in a linkage-specific manner.

A ubiquitinating enzyme has a regulatory function in the ubiquitin-mediated regulation in vivo, and DUB activity is highly regulated in dynamic environments through protein-protein interaction, posttranslational modification, and re-localization (Clague et al, 2019; Meyer et al, 2020). In yeast, physiological functions of Ubp3 have been reported, including the replicative life span (Oling et al, 2014), oxidative stress response (Silva et al, 2015; Simões et al, 2022), and degradation of stalled RNA polymerase after UV irradiation (Kvint et al, 2008; Milligan et al, 2017). Ubp3 is also involved in transcriptional activation upon osmotic stress (Solé et al, 2011) and translation (Takehara et al, 2021), stress granule formation (Nostramo and Herman, 2016; Nostramo et al, 2016). Ubp3 and its cofactors Cdc48 and Ufd3 are required for ribophagy, starvation-induced selective autophagy (Kraft et al, 2008; Ossareh-Nazari et al, 2010), and ubiquitination by the Ltn1 E3 ligase protects the 60S ribosomal subunit (Ossareh-Nazari et al, 2014), suggesting that a crucial role of the cycle ubiquitination of ribosome in ribophagy. In mammals, the ribosome stalling induced by genotoxic stress such as ultraviolet light-B or 4-nitroquinoline 1-oxide stress recruits ASCC3, however, ASCC3 failed to resolve them, resulting in a prolonged cell-cycle arrest through the ZAK-p38MAPK signaling axis (Stoneley et al, 2022). The collided ribosomes with the K63-linked polyubiquitin chain on uS10 are a primary target for the RQT complex, and the removal of the K48/K63-mixed polyubiquitin chain on uS10 could have a regulatory function in mammals. We also assume that a putative E3 ligase(s) may modulate the linkage of the polyubiquitin chain on uS10 to affect the splitting efficiency of the collided ribosomes. Regardless of the regulation by E3 ligase or deubiquitinates, further analysis of the length of the ubiquitin chain on the ribosome proteins formed by endogenous stalling sequences, translation inhibitors, and genotoxic stresses may contribute to the understanding of the fate of the stalled ribosomes.

# Methods

### Reagents and tools table

| Reagent/resource | Reference or source | Identifier or catalog number |
|---|---|---|
| **Experimental models** | | |
| W303-1a (*S. cerevisiae*) | Inada lab and this study | Appendix Table S1 |
| BY4741 (*S. cerevisiae*) | Open Biosystems | Appendix Table S1 |
| **Recombinant DNA** | | |
| Yeast plasmids | Inada lab, this study | Appendix Table S2 |
| *pOPINB-AMSH** | Addgene | Cat #66712 |
| *pOPINB-OTUB1** | Addgene | Cat #65441 |
| **Antibodies** | | |
| Anti-HA-Peroxidase from Roche | Roche | Cat #12013819001 |
| Anti-FLAG M2 | Sigma-Aldrich | Cat # F1804 |
| GFP(B-2) mouse monoclonal IgG2a | Santa Cruz Biotechnology | Cat# sc-9996 |
| Anti-eEF2 antibody | Inada lab | N/A |
| Anti-eEF3 antibody | Inada lab | N/A |
| Mouse Anti-Viral V5-TAG monoclonal antibody, Unconjugated, Clone SV5-Pk1 | Bio-Rad | Cat #MCA1360 |
| Anti-Ubiquitin (P4D1) HRP | Santa Cruz Biotechnology | Cat #sc-0817 |
| Anti-K48 Ubiquitin antibody | Cell Signaling Technology | Cat #8081 |
| Anti-K63 Ubiquitin antibody | EMD Millipore | Cat #05-1308 |
| ECL Anti-mouse IgG, horseradish Peroxidase (HRP)-conjugated secondary antibodies | GE Healthcare | Cat #NA931V |
| ECL Anti-rabbit IgG, horseradish Peroxidase (HRP)-conjugated secondary antibodies | GE Healthcare | Cat #NA934V |
| **Oligonucleotides and other sequence-based reagents** | | |
| **Chemicals, enzymes, and other reagents** | | |
| Anisomycin | MedChemExpress | HY-18982 |
| Cycloheximide | Nacalai tesque | Cat# 06741-04 |
| cOmplete™, Mini, EDTA-free Protease Inhibitor Cocktail | Roche | Cat# 11836170001 |
| MG132 | MedChemExpress | HY-13259 |
| PR-619 | UBPBio | F2111 |
| DYKDDDDK peptide | GenScript | N/A |
| MNase | Takara | Cat # 2910 A |
| AMSH* | This study | N/A |
| OTUB1* | This study | N/A |
| PreScission Protease | GE Healthcare | Cat# 27084301 |

| Reagent/resource | Reference or source | Identifier or catalog number |
|---|---|---|
| K48-Ub4 | UBPBio | Cat #D1300 |
| K63-Ub4 | UBPBio | Cat #D2300 |
| Recombinant Human Tetra-Ub/Ub4 WT Chains (K63/K48/K63) | R&D Systems | UCM-210-025 |
| Ubp2 WT | This study | N/A |
| Ubp2 C745S | This study | N/A |
| Ubp3 WT | This study | N/A |
| Ubp3 C496A | This study | N/A |
| ImmunoStar LD | Fujifilm | Cat# 290-69904 |
| mMESSAGE mMACHINE™ T7 Transcription Kit | Thermo Fisher Scientific | AM1344 |
| **Software** | | |
| ImageJ | https://imagej.net/ij/index.html | |
| RStudio | https://rstudio.com/ | |
| PinPoint software | Thermo Fisher Scientific | |
| **Other** | | |
| ImageQuant LAS4000 mini | GE Healthcare | |
| Gradient Master | BioComp | |
| Piston Gradient Fractionator | BioComp | |
| ATTO Biomini UV-monitor | ATTO | |
| ATTO digital mini-recorder | ATTO | |
| 0.5 mm Zirconia/Silica Beads | BioSpec Products | Cat # 11079105z |
| Anti-DYKDDDDK tag, Antibody Beads | Wako | Cat# 016-22784 |
| Ni-NTA agarose | QIAGEN | Cat #30210 |
| Glutathione Sepharose 4B | GE Healthcare | Cat# 17-1756-05 |
| Dynabeads M270 Epoxy | Thermo Fisher Scientific | 14302D |
| IgG from rabbit serum | SIgma-Aldrich | I5006 |
| Silver stain KANTO III | KANTO CHEMICAL | Cat#37937-96 |

## Yeast strains and plasmids

Yeast strains W303-1a and its derivatives used in this study are listed in Appendix Table S1. Yeast knock-out library strains (BY4741) (Open Biosystems) used in the deubiquitinating enzyme screening are indicated in Fig. EV1. Gene disruption and C-terminal tagging were performed as previously described (Janke et al, 2004; Longtine et al, 1998). Yeast plasmids and their derivatives used in this study are listed in Appendix Table S2.

## Yeast culture and media

All yeast cells were grown in YPD or synthetic complete (SC) medium with 2% glucose at 30 °C and harvested at log phase ($OD_{600}$ of 0.5–0.7) by centrifugation and discarding medium unless otherwise noted. For polysome analysis, yeast cells were cultured at 30 °C and treated with 0.1 mg/mL cycloheximide (Nacalai Tesque) for 5 min on ice bath before harvesting. In all, 0.5 mM MG132 was treated for 2 h before harvesting.

## Cell lysis for protein preparation

The yeast cell pellet in a 1.5-mL tube (on ice) was resuspended with 500 μL ice-cold lysis buffer (20 mM Tris-HCl pH 7.5, 150 mM NaCl, 1.8 mM $MgCl_2$, 0.5% NP-40, 1 mM PMSF, 1 mM DTT, 20 μM MG132, 1 tablet/10 mL cOmplete mini EDTA-free (Roche)) and transferred to a new 1.5-mL tube containing 50 μL of 0.5 mm Zirconia/Slica Beads (BioSpec Products). The cells were vortexed for 10 s six times, and the supernatant was transferred to a new 1.5-mL tube. After centrifugation of lysates (14,000 rpm, 10 min, 4 °C), the 30 μL of supernatant was transferred to new 1.5-mL tube with 10 μL of 4×SDS sample buffer (200 mM Tris-HCl pH 6.8, 8% SDS, 40% glycerol, 100 mM DTT, 0.04% bromophenol blue; 150 μL/$6OD_{600}$) and heated at 88 °C for 10 min.

## Trichloroacetic acid (TCA) precipitation for protein preparation

The yeast cell pellet in a 1.5 mL tube (on ice) was resuspended with 500 μL ice-cold TCA buffer (20 mM Tris-HCl pH 8.0, 50 mM $NH_4OAc$, 2 mM EDTA, and 1 mM PMSF) and transferred to a new 1.5 mL tube containing 500 μL of 20% TCA and 500 μL of 0.5 mm Zirconia/Silica Beads (BioSpec Products). The cells were vortexed for 30 s three times, and the supernatant was transferred to a new 1.5-mL tube. Another 500 μL ice-cold TCA buffer was added to a beads-containing 1.5-mL tube, vortexed for 30 s, and then the supernatant was transferred to a 1.5-mL tube. After centrifugation of lysates (14,000 rpm, 10 min, 4 °C), the supernatant was discarded, and the pellet was dissolved in SDS sample buffer (125 mM Tris-HCl, 4% SDS, 20% glycerol, 100 mM DTT, 0.01% bromophenol blue; 150 μL/$6OD_{600}$) and heated at 100 °C for 10 min followed by centrifugation at 13,000 rpm for 10 min, room temperature.

## Sucrose density gradient centrifugation (SDG)

The cell pellet was frozen and ground in liquid nitrogen using a mortar. The cell powder was resuspended with lysis buffer (20 mM HEPES–KOH pH 7.4, 100 mM KOAc, 2 mM $Mg(OAc)_2$, 0.5 mM DTT, 1 mM PMSF and one tablet/10 mL complete mini EDTA-free (Roche)) to prepare the crude extracts. 50 μM PR-619, a DUB inhibitor, was added to the buffer only when analyzing ubiquitination. For the polysome analysis of Ubp3, the cell powder was resuspended with LB100 (50 mM Tris-HCl pH 7.5, 100 mM NaCl, 10 mM $MgCl_2$, 1 mM DTT, 1 mM PMSF, and one tablet/10 mL complete mini EDTA-free (Roche)) and 300 ng of RNA in crude extracts was treated with 2.5 mM $CaCl_2$ and 20 U of MNase (Takara) at 37 °C for 5 min and then quenched with EGTA at a final concentration of 2.5 mM. Sucrose gradients (10–50% sucrose

in 10 mM Tris-acetate pH 7.4, 70 mM $NH_4OAc$, and 4 mM $Mg(OAc)_2$) were prepared in 25 × 89 mm polyallomer tubes (Beckman Coulter) using a Gradient Master (BioComp). Crude extracts (the equivalent of 50 A260 units) were layered on top of the sucrose gradients and centrifuged at 150,000 × $g$ in a P28S rotor for 3.0 h at 4 °C or 197,568 × $g$ in a SW40 rotor for 1.5 h at 4 °C. The gradients were then fractionated with BioComp Piston Gradient Fractionator. The polysome profiles were generated by continuous absorbance measurement at 254 nm using a single path UV-1 optical unit (ATTO Biomini UV-monitor) connected to a chart recorder (ATTO digital mini-recorder). For the western blots, 360 μL of each fraction was mixed with 40 μL of 100% TCA and incubated for 15 min at 4 °C. After centrifugation (14,000 rpm, 15 min, 4 °C), the supernatant was removed, and the pellet was washed by acetone and dissolved in 40 μL of SDS sample buffer (125 mM Tris-HCl, 4% SDS, 20% glycerol, 100 mM DTT, 0.01% bromophenol blue).

## Electrophoresis, silver stain

Protein samples were separated by SDS-PAGE or Neu-PAGE and analyzed by silver staining or transferred onto PVDF membranes (Immobilon-P, Merck Millipore, MA). Membranes were blocked with 5% skim milk in PBST (10 mM $Na_2HPO_4$/$NaH_2PO_4$ pH 7.5, 0.9% NaCl, and 0.1% Tween 20), incubated with primary antibodies for 1 h at room temperature, washed three times in PBST, and incubated with horseradish peroxidase (HRP)-conjugated secondary antibodies for 1 h at room temperature. For HA-tagged protein detection, the membrane was incubated with HRP-conjugated antibodies. After washing with PBST three times, chemiluminescence was detected with the LAS4000 system (GE Healthcare). Primary antibodies used for western blotting were as follows: anti-HA-peroxidase (Roche); anti-FLAG M2 antibody (Sigma); anti-GFP (Santa Cruz Biotechnology); anti-K48 Ubiquitin antibody (Cell Signaling Technology); anti-K63 Ubiquitin antibody (EMD Millipore). Silver stain was performed by the silver stain KANTO III.

## Two-steps affinity purification by uL23-FLAG containing ribosome and 6His-tagged uS10

The ubiquitylated ribosomal proteins were purified by a two-step affinity purification method using uL23-FLAG and 6His-tagged uS10 expressing plasmid. 2 L of yeast cells harboring p*Hel2-V5* and p*uL23-FLAG* were cultured in SC containing 2% glucose. Ground yeast pellet was resuspended in ice-cold LB150 (50 mM Tris-HCl pH 7.5, 150 mM KCl, 10 mM $Mg(OAc)_2$, 0.05% NP-40, 2 mM 2-mercaptoethanol, 1 mM PMSF, 50 μM PR-619) containing cOmplete Mini EDTA-free (Roche)(1 tablet/10 mL), centrifuged at 10,000 × $g$, 4 °C for 10 min followed by thorough centrifugation of supernatant at 40,000 × $g$, 4 °C for 30 min to obtain clear lysate. To purify the FLAG-tagged ribosomes, the lysate was incubated at 4 °C with 200 μL of pre-equilibrated anti-DYKDDDDK tag antibody beads (Wako) for 1 h. After washing steps by batch with LB150 for seven times, ribosomes were eluted from beads by incubation with 600 μL of LB150 containing 100 μg/mL FLAG peptide (GenScript) at 4 °C for 1 h. The eluted ribosomes were incubated at denaturation conditions (50 mM Tris-HCl pH 7.5, 300 mM NaCl, 20 mM Imidazole. 6 M Guanidine-HCl, 2 mM 2-mercaptoethanol,

0.01% NP-40). To purify uS10, the denatured ribosomes were incubated at 4 °C with 50 μL of pre-equilibrated Ni-NTA agarose (QIAGEN) for 1 h. After washing steps by batch with Wash Buffer 1 (50 mM Tris-HCl pH 7.5, 300 mM NaCl, 20 mM Imidazole. 6 M Guanidine-HCl, 5 mM 2-mercaptoethanol, 0.05% NP-40) for two times and Wash Buffer 2 (50 mM Tris-HCl pH 7.5, 100 mM NaCl, 10 mM $MgCl_2$, 10 mM Imidazole. 2 mM 2-mercaptoethanol, 0.01% NP-40) for two times, uS10 was eluted from beads by incubation with 600 μL of Elution Buffer (50 mM Tris-HCl pH 7.5, 100 mM NaCl, 10 mM $MgCl_2$, 300 mM Imidazole. 2 mM 2-mercaptoethanol, 0.01% NP-40) at 4 °C for 1 h. Eluted uS10 were concentrated by TCA precipitation method and dissolved with the sample buffer.

## Purification of recombinant proteins of OTUB1* and AMSH*

Recombinant His6-OTUB1* and His6-AMSH* was purified from *E. coli* Rossetta-gami 2 (DE3) (Novagen). *E. coli* cells were grown in 200 ml of 2xTY medium supplemented with 50 μg/mL kanamycin and 35 μg/mL chloramphenicol at 37 °C to $OD_{600} = 0.3$, cooled to 18 °C, and induced with 0.4 mM isopropyl thio-β-galactopyranoside (IPTG) for 16 h before being harvested. *E. coli* cells were lysed by ultrasonic wave with 10 mL of ice-cold lysis buffer (50 mM Tris pH 7.4, 150 mM NaCl, 2 mM β-mercaptoethanol) containing cOmplete Mini EDTA-free (Roche) (1 tablet/10 mL). The lysate was centrifuged at 10,000 × *g*, 4 °C for 10 min followed by thorough centrifugation of the supernatant at 40,000 × *g*, 4 °C for 30 min to obtain a clear lysate. The clear lysate was incubated with 200 μL of Ni-NTA agarose (QIAGEN) pre-equilibrated with lysis buffer, followed by washing with lysis buffer five times. OTUB1* and AMSH* moiety was eluted from beads using 200 μL of elution buffer (50 mM Tris pH 7.4, 150 mM NaCl, 2 mM β-mercaptoethanol, 8 μL of (GE Healthcare) at 4 °C for 16 h. PreScission protease was removed by Glutathione Sepharose 4B (GE Healthcare).

## Purification of Ubp2 and Ubp3 protein

Ubp2-3xFLAG and Ubp3-3xFLAG were purified from 1 L of SC 2% glucose culture of yeast cell harboring p*GPDp-Ubp2-3xFLAG-CYCt* or p*GPDp-Ubp3-3xFLAG-CYCt*. Ground yeast pellet was resuspended in ice-cold LB500 (50 mM Tris-HCl pH 7.5, 500 mM NaCl, 10 mM $MgCl_2$, 0.01% NP-40, 1 mM DTT, 1 mM PMSF) containing cOmplete Mini EDTA-free (Roche) (1 tablet/10 mL), centrifuged at 10,000 × *g*, 4 °C for 10 min followed by thorough centrifuge of supernatant at 40,000 × *g*, 4 °C for 30 min to obtain clear lysate. To purify Ubp2-3xFLAG and Ubp3-3xFLAG, the lysate was incubated at 4 °C with 100 μL of pre-equilibrated anti-DYKDDDDK tag antibody beads (Wako) for 1 h. After washing steps by batch with LB500 for five times, LB400 for one time, LB300 for one time, LB200 for one time and LB100 w/o detergent for three times, Ubp2-3xFLAG and Ubp3-3xFLAG were eluted from beads by 100 μL of Elution Buffer at 4 °C for 1 h.

## UbiCRest

Deubiquitylation reactions were performed in reaction buffer (50 mM Tris-HCl pH 7.5, 100 mM NaCl, 10 mM $MgCl_2$, 2 mM β-mercaptoethanol) contained 25–100 nM AMSH*, 50–200 nM OTUB1* 1000–20,000 ng Ribosome, 100 ng K48-Ub4 (UBPBio),

100 ng K63-Ub4 (UBPBio). To stop the ubiquitination reaction, 4× Laemmli sample buffer was added to the reaction tube. To purify 6His-tagged uS10, the method described above was performed.

## In vitro deubiquitination assay by Ubp2 and Ubp3

The total cell lysate was separated by sucrose density gradient. The 400 μL of 40S fraction or polysome fraction was transferred to a 1.5-mL tube. Deubiquitinase or a catalytic mutant was added to a 1.5-mL tube. Reaction was mixed and incubated at 30 °C for 1 h. To stop the ubiquitylation reaction, transfer 90 μL of reaction solution to a new 1.5-mL tube and add 10 μL of 100% TCA. After centrifugation (14,000 rpm, 10 min, 4 °C), the supernatant was removed, and the pellet was washed with acetone and dissolved in 10 μL of SDS sample buffer (125 mM Tris-HCl, 4% SDS, 20% glycerol, 100 mM DTT, 0.01% bromophenol blue).

## Pull-down assay of the RQT complex with the tetraubiquitin chain

The yeast strain overexpressing RQT factors (*SLH1-FTP, CUE3-Flag, RQT4-Flag*) were cultured in synthetic complete medium. The harvested cell pellet was frozen in liquid nitrogen and then ground in liquid nitrogen using a mortar. The cell powder was resuspended with RQT-R buffer (50 mM Tris pH 7.5, 100 mM NaCl, 2.5 mM $MgCl_2$, 0.1% NP-40, 10% glycerol, 100 mM L-arginine, 1 mM DTT, 10 μM $ZnCl_2$, and 1 mM PMSF) to prepare the lysate. The lysate was centrifuged at 39,000× *g* for 30 min at 4 °C, and the supernatant fraction was used for the purification step. The RQT complex was immobilized with IgG beads (Cytiva). The immobilized RQT complex was mixed with K48-, K63-, K48/K63-mixed linkage tetraubiquitin (purchased from UBPBio or R&D systems) in RQT-R buffer and incubated for 15 min at 23 °C. After the reaction, the immobilized RQT complex with IgG beads was washed with RQT-R buffer and eluted with LDS-sample buffer (Thermo Fischer). The elution was separated by 10% Nu-PAGE and analyzed by CBB staining or immunoblotting using an anti-ubiquitin antibody (Santa Cruz Bio.).

## In vitro translation of SDD1 mRNA and Purification of RNCs on the SDD1 mRNA

*His-SDD1 reporter* mRNA was produced using the mMessage mMachine Kit (Thermo Fischer) and used in a yeast cell-free translation extract. This yeast translation extract was prepared, and in vitro translation was performed as described previously (Waters and Blobel, 1986). The cells were grown in YPD medium to an $OD_{600}$ of 1.5–2.0; washed with water and 1% KCl; and finally incubated with 10 mM DTT in 100 mM Tris, pH 8.0 for 15 min at room temperature. To generate spheroplasts, 2.08 mg zymolyase per 1 g of cell pellet was added in YPD/1 M sorbitol and incubated for 75 min at 30 °C. Spheroplasts were then washed three times with YPD/1 M sorbitol and once with 1 M sorbitol, and lysed as described previously (Waters and Blobel, 1986) with a douncer in lysis buffer [20 mM HEPES pH 7.5, 100 mM KOAc, 2 mM Mg(OAc)$_2$, 10% Glycerol, 1 mM DTT, 0.5 mM PMSF, and complete EDTA-free protease inhibitors (Cytiva)]. From the lysate, an S100 fraction was obtained by low-speed centrifugation followed

by ultracentrifugation of the supernatant. The S100 was passed through a PD10 column (Cytiva). In vitro translation was performed at 17 °C for 60 min using a great excess of template mRNA (20 µg per 200 µl of extract) to prevent degradation of the resulting stalled ribosomes by endogenous response factors.

The stalled RNCs on the *His-SDD1* mRNA were affinity-purified using the His-tag on the nascent polypeptide chain. After in vitro translation reaction, the extract was applied to Dynabeads™ (Thermo Fischer) for His-tag isolation and pull-down for 5 min at 4 °C. The beads were washed with lysis buffer 500 (50 mM HEPES pH 6.8, 500 mM KOAc, 10 mM Mg(OAc)₂, 0.01% NP-40, and 5 mM β-mercaptoethanol) and eluted in elution buffer (50 mM HEPES pH 7.5, 100 mM KOAc, 2.5 mM Mg(OAc)₂, 0.01% NP-40, and 5 mM β-mercaptoethanol) containing 300 mM imidazole.

### In vitro splitting assay

The purified RNCs (100 nM ribosome) containing 100 nM Rqc2 were incubated with or without 10 nM RQT complex and 1 mM ATP in reaction buffer (50 mM HEPES, pH 7.4, 100 mM KOAc, 2.5 mM Mg(OAc)₂, 0.01% NP-40, 5 mM β-mercaptoethanol, and 0.2 U/µl SUPERase In RNase inhibitor) for 45 min at 25 °C. After incubation, ribosomal fractions were separated by sucrose density gradient centrifugation. The RNCs were monitored by measuring UV absorbance at 254 nm, and the ubiquitinated uS10 in each fraction was detected by immunoblotting using an anti-HA antibody.

### Western blotting

Western blotting was performed as previously described (Sugiyama et al, 2019). Briefly, protein samples were heated at 88 °C for the detection of ubiquitinated uS10 and otherwise at 95 °C for 10 min, followed by centrifugation at 16,000 × g for 10 min at room temperature before loading to gels. Proteins were separated by SDS-PAGE and transferred onto a PVDF membrane (Immobilon-P, Millipore), blocked with 5% skim milk, and incubated with antibodies. The following primary and secondary antibodies were used: Anti-HA-Peroxidase from Roche (# 12013819001, RRID: AB 390917); Anti-FLAG M2 from Sigma-Aldrich (# F1804); Anti-eEF3 antibody (Lab stock); Mouse Anti-Viral V5-TAG monoclonal antibody, Unoconjugated, Clone SV5-Pk1 from Bio-Rad (# MCA1360, AB_322378); Anti-Ubiquitin (P4D1) HRP from Santa Cruz Biotechnology (# sc-0817, RRID:AB_628423); Anti-K48 Ubiquitin antibody from Cell Signaling Technology (# 8081; RRID:AB_1587580); Anti-K63 Ubiquitin antibody from EMD Millipore (# 05-1308; RRID:AB_10859893); ECL Anti-mouse (# NA931V) and Anti-rabbit (# NA934V) IgG, horse-radish Peroxidase (HRP)-conjugated secondary antibodies from GE Healthcare. The membrane was then washed and reacted with the homemade ImmunoStar solution. Chemiluminescence was detected by ImageQuant LAS4000 mini (GE Healthcare).

### Mass spectrometric analysis

For liquid chromatography–tandem mass spectrometry (LC-MS/MS) analyses shown in Fig. 3, in-gel digestion was performed as previously described with minor modifications (Tsuchiya et al, 2017). After the separation of analytes using SDS-PAGE, the gel pieces were washed in 50 mM AMBC/30% acetonitrile (ACN) and then with 50 mM AMBC/50% ACN for 2 h. The gel pieces were then dehydrated in 100% ACN, and proteins were digested at 37 °C for 16 h with 20 ng/µL sequence-grade trypsin (Promega) in 50 mM AMBC/5% ACN, pH 8.0. The digested peptides were extracted three times with 0.1% trifluoroacetic acid (TFA)/80% ACN. A mixture of AQUA peptides (25 fmol/injection) was added to the extracted peptides, and the concentrated peptides were diluted with 20 µL of 0.1% TFA containing 0.05% H₂O₂. For quantification of branched and unbranched K48/K63 linkages, the peptides were diluted with 50 mM AMBC for neutralization, and incubated in a 20 µL reaction with 2 ng/µl recombinant glutaminyl-peptide cyclotransferase (QPCT) (R&D Systems) for 6 h at 37 °C (Ohtake et al, 2016). Subsequently, the peptides were purified using C18 tips (Thermo Scientific) and GL-GC tips (GL Science).

For ubiquitin linkage analysis shown in Fig. 5, purified uS10-3HA-6His were denatured with 1% SDS, reduced with 10 mM dithiothreitol (DTT) at 70 °C for 10 min, and alkylated with 15 mM iodoacetamide (IAA) for another 15 min at room temperature in the dark, followed by on-beads tryptic digestion based on a single-pot solid-phase-enhanced sample preparation (SP3) method (Müller et al, 2020) using a KingFisher APEX equipped with a 96 Combi head (Thermo Fisher Scientific). Analytes were incubated in 200 µL of 50% EtOH and conjugated to 40 µg of an equal mixture of hydrophobic and hydrophilic SeraMag SpeedBead carboxylate-modified magnetic particles (Cytiva). After desalting thrice with 400 µL of 80% EtOH, the magnetic beads were subjected to enzymatic digestion in 100 µL of 5 ng/µL Trypsin Gold (Promega) and 50 mM AMBC for 3.5 h at 37 °C and rinsed out with 100 µL of pure water. The trypsin and rinsing solutions were combined, and a mixture of AQUA peptides (25 fmol/injection) was added to the extracted peptides, followed by concentration using a speed-vac. The peptide samples were diluted with 20 µL of 0.1% TFA containing 0.05% H₂O₂.

For LC-MS/MS analysis, an Easy nLC 1200 (Thermo Fisher Scientific) was connected inline to an Orbitrap Fusion LUMOS (Thermo Fisher Scientific) with a nanoelectrospray ion source (Thermo Fisher Scientific) or a Dream Spray ING ion source (AMR). Peptides were separated on a C18 analytical column (IonOpticks, Aurora Series Emitter Column, AUR2-25075C18A 25 cm × 75 µm 1.6 µm or AUR3-15075C18 15 cm × 75 µm 1.7 µm with a nanoZero fitting) using a 55 min or a 31 min gradient (solvent A, 0.1% FA; and solvent B, 80% ACN/0.1% FA). Targeted acquisition of MS/MS spectra (parallel reaction monitoring) was performed using an Orbitrap Fusion LUMOS instrument operated in targeted MS/MS mode using the Xcalibur software. The peptides were fragmented using higher-energy collisional dissociation (normalized collision energy of 28), and the fragment ions were detected using an Orbitrap. Data were processed using the PinPoint software (Thermo Fisher Scientific), and peptide abundance was calculated based on the integrated area under the curve for the selected fragment ions.

## Data availability

Proteomics data have been deposited to ProteomeXchange via the PRIDE database with the accession code PXD052346.

The source data of this paper are collected in the following database record: biostudies:S-SCDT-10_1038-S44318-025-00568-0.

## Peer review information

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

## Acknowledgements

This work was supported by AMED (JP23gm1110010, JP223fa627001 to TI; 24gm1410007h0004 to F.O.), MEXT/JSPS KAKENHI (Grant Numbers JP19H05281, 21H05277, 22H00401, 25H00007 to TI, 25H01442 and 25K02221 to YM, 18H05498 to YS and FO; JP23H04922 to F.O.), Research grants from Takeda Science Foundation (TI), and JST PREST Grant Number JPMJPR21EE and JPMJPR2488 (YM).

## Author contributions

**Shota Tomomatsu**: Resources; Data curation; Formal analysis; Validation; Investigation; Visualization; Writing—original draft. **Yoshitaka Matsuo**: Conceptualization; Resources; Data curation; Formal analysis; Supervision; Funding acquisition; Validation; Investigation; Visualization; Methodology; Writing—original draft; Writing—review and editing. **Fumiaki Ohtake**: Supervision; Validation; Investigation; Visualization; Methodology. **Takuya Tomita**: Validation; Investigation; Visualization; Methodology. **Yasushi Saeki**: Supervision; Validation; Methodology. **Toshifumi Inada**: Conceptualization; Formal analysis; Supervision; Funding acquisition; Validation; Visualization; Methodology; Writing—original draft; Project administration; Writing—review and editing.

Source data underlying figure panels in this paper may have individual authorship assigned. Where available, figure panel/source data authorship is listed in the following database record: biostudies:S-SCDT-10_1038-S44318-025-00568-0.

## Disclosure and competing interests statement

The authors declare no competing interests.

# Expanded View Figures

**A**

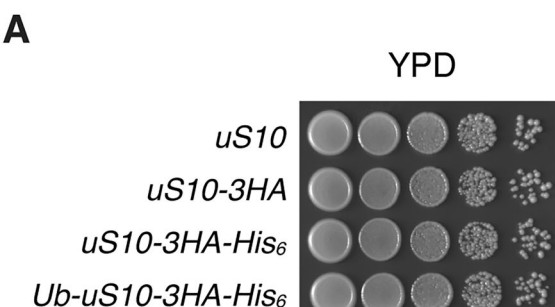

**B**

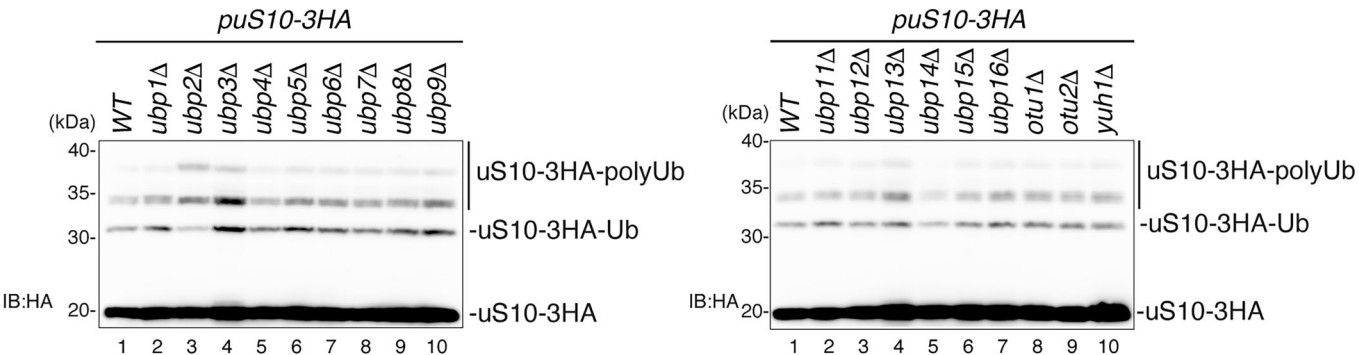

**C**

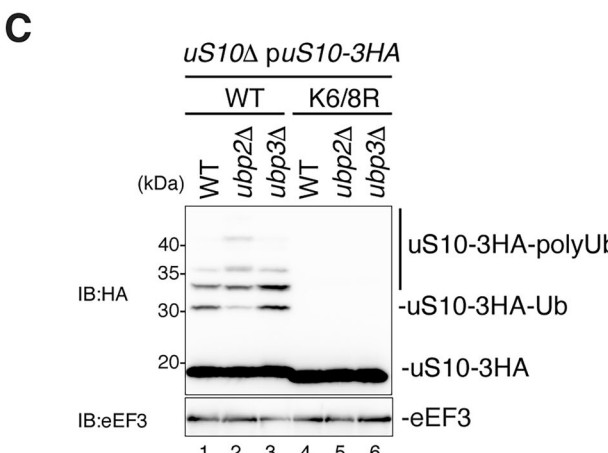

**Figure EV1.   Ubp2 and Ubp3 are involved in the deubiquitination of uS10.**

(A) Spot assay of the indicated uS10 tagged or untagged strains. The indicating cells: *uS10Δ* expressing uS10, uS10-3HA, uS10-3HA-His$_6$, or Ub-uS10-3HA-His$_6$ from plasmids pKI124, pKI236, pST051, and pST158, respectively, diluted to OD$_{600}$ = 0.3 and 10-fold serial dilutions were spotted and incubated at 30°C for two days. (B) Genetic screening to identify the deubiquitinating enzymes for uS10. Protein samples prepared from the indicated mutant cells: wild-type, *ubp1Δ, ubp2Δ, ubp3Δ, ubp4Δ, ubp5Δ, ubp6Δ, ubp7Δ, ubp8Δ, ubp9Δ, ubp11Δ, ubp12Δ, ubp13Δ, ubp14Δ, ubp15Δ, ubp16Δ, otu1Δ, otu2Δ, and yuh1Δ*, expressing uS10-3HA from plasmid pST001 were subjected to immunoblotting using an anti-HA antibody. Total proteins used for the immunoblotting were prepared by Cell lysis method. (C) The analysis of the ubiquitination of the uS10-K6/8 R mutant in the *ubp2Δ* and *ubp3Δ* strains. The ubiquitin level of uS10-3HA or uS10K6/8R-3HA derived from plasmid pKI236 or pKI237 in the *uS10Δ, uS10Δubp2Δ, uS10Δubp3Δ* mutants was detected with immunoblotting using an anti-HA antibody.

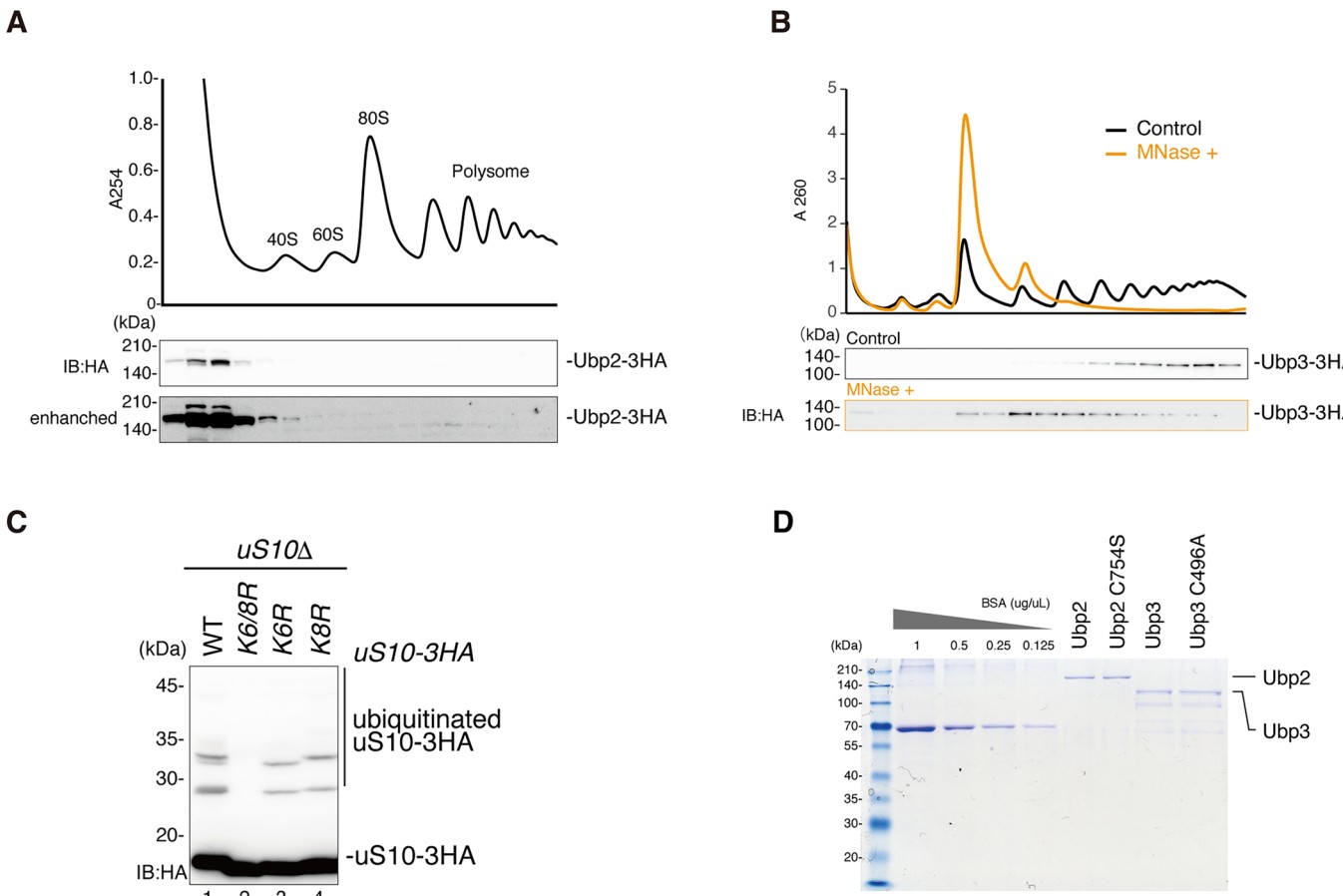

**Figure EV2. Ubp2 and Ubp3 are associated with 40S subunit and polysome, respectively.**

(A, B) The total lysates derived from 3 x HA genomic tagging Ubp2 (A) or Ubp3 (B) expressing cells were subjected to the sucrose density gradient and sedimented through by ultracentrifugation. For Ubp3, lysates were prepared either with or without micrococcal nuclease (MNase) treatment. Ubp2-3HA and Ubp3-3HA in each fraction were detected by immunoblotting using an anti-HA antibody. (C) The polyubiquitination of uS10 in K6R and K8R mutant of uS10. The 3 x HA-tagged uS10, uS10-K6/8 R, uS10-K6R or K8R was expressed from plasmids pKI237, pKI238, and pKI239, respectively, in the *uS10Δ* cells and detected the HA-tagged uS10 by immunoblotting using an anti-HA antibody. Total proteins used for immunoblotting were prepared by Cell lysis method. (D) CBB stain of purified Ubp2, Ubp2 C745S, Ubp3, Ubp3C496A.

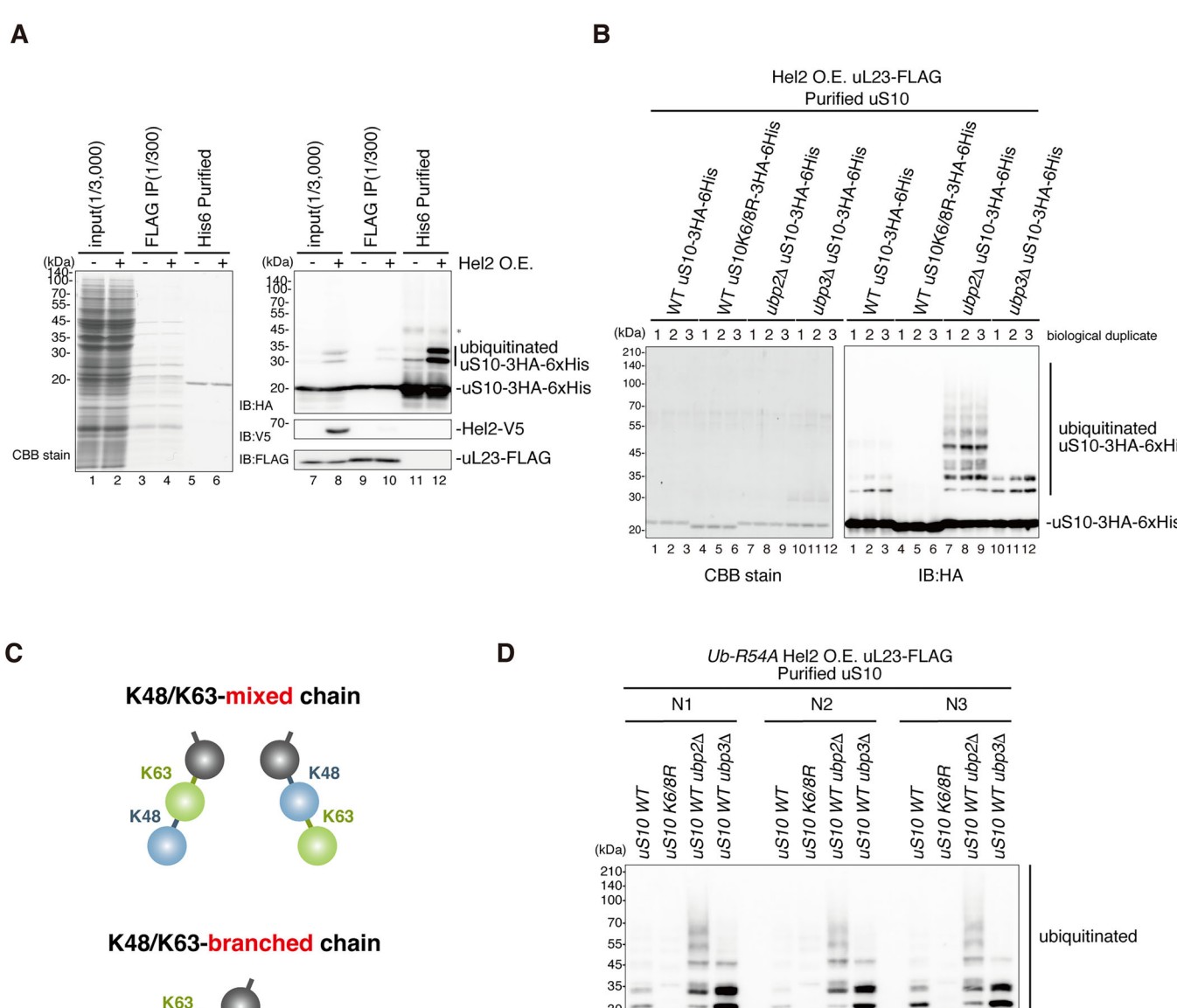

**Figure EV3. The polyubiquitin chains formed on uS10 are mainly K63- and K48-linkage.**

(A) The two-step purification of uS10-3HA-His$_6$. The uS10-3HA-His$_6$, uL23-FLAG, and Hel2-V5 were expressed from plasmids pST051, pKI191, and pST069 in *uS10Δ* cells. The ubiquitinated uS10-3HA-His$_6$ were purified by two-step affinity purification. The purified samples were stained with CBB (Left) or subjected to Immunoblotting using an anti-HA antibody, anti-V5, and anti-FLAG antibody (Right). (B) The sample preparation for Absolute quantification analysis of Ub chains. The uS10-3HA-His$_6$ or uS10-K6/8R-3HA-His$_6$ together with uL23-FLAG, and Hel2-V5 were expressed from plasmids pST051, pST052, pKI191, and pST069 in the following strains: *uS10Δ, uS10Δubp2Δ, uS10Δubp3Δ*. The ubiquitinated uS10-3HA-His$_6$ were purified by two-step affinity purification. The purified samples were stained with CBB (Left) or subjected to Immunoblotting using an anti-HA antibody (Right). (C) Schematic drawing of K48/K63 mixed or branched ubiquitin chain architectures. (D) The sample preparation for Absolute quantification analysis of Ub chains. The uS10-3HA-His$_6$ or uS10-K6/8R-3HA-His$_6$ together with uL23-FLAG, Hel2-V5, and Ub-R54A were expressed from plasmids pST051, pST052, pKI191, pST069, and pUB100-R54A in the following strains: *ubi1-4ΔuS10Δ, ubi1-4ΔuS10Δubp2Δ, ubi1-4ΔuS10Δubp3Δ*. The ubiquitinated uS10-3HA-His$_6$ were purified by two-step affinity purification. The purified samples were subjected to Immunoblotting using an anti-HA antibody.

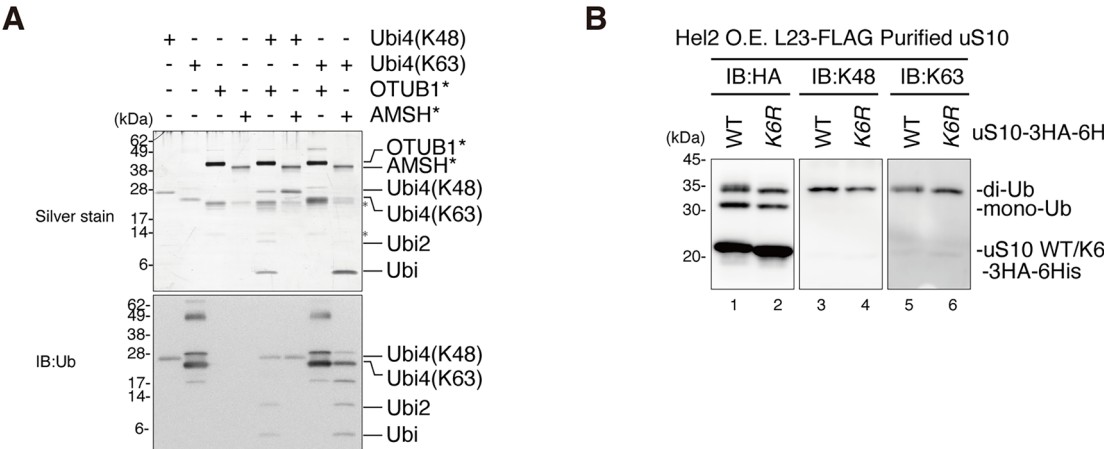

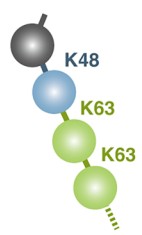

**Figure EV4. The K48- and K63-linked di-ubiquitin chains are formed on uS10-K6R.**

(A) K48-linked or K63-linked tetraubiquitin chains were reacted with AMSH* (K63-linkage-specific deubiquitinase) and OTUB1* (K48-linkage specific deubiquitinase). The protein samples were separated by 15% Nu-PAGE and detected by silver staining (top panel) or immunoblotting using an anti-ubiquitin antibody (bottom panel). (B) Both K48- and K63-linked polyubiquitin chain was formed on uS10. The uS10-WT-3HA-His$_6$ or uS10-K6R-3HA-His$_6$ together with uL23-FLAG and Hel2-V5 were expressed from plasmids pST051, pST137, pKI191, and pST069, respectively, in *uS10Δ* cells. The uS10-WT-3HA-His$_6$ or uS10-K6R-3HA-His$_6$ were purified by two-step affinity purification. Immunoblotting of purified samples using an anti-HA antibody, K48-linkage-specific anti-ubiquitin antibody, and K63-linkage-specific anti-ubiquitin antibody. (C) Schematic drawing of the architecture of K48/K63 mixed ubiquitin chain via K48-linked di-ubiquitination.

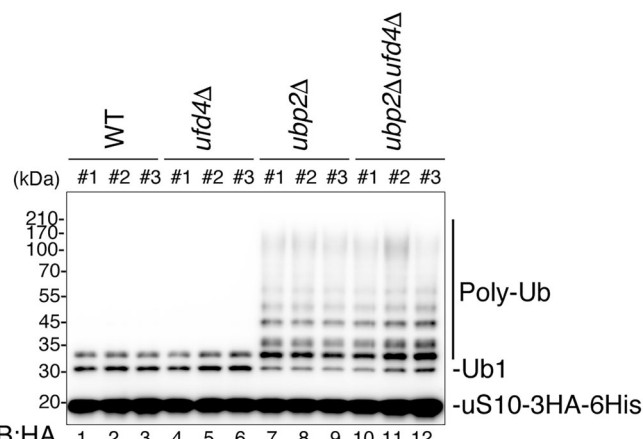

**Figure EV5.  The sample preparation for Absolute quantification analysis of Ub chains in the *ufd4Δ* mutant.**

The uS10-3HA-His₆, uL23-FLAG, and Hel2-V5 were expressed from plasmids pST051, pKI191, and pST069 in the following strains: *uS10Δ, uS10Δufd4Δ, uS10Δubp2Δ, uS10Δufd4Δ*. The ubiquitinated uS10-3HA-His₆ were purified by two-step affinity purification. The purified samples were subjected to Immunoblotting using an anti-HA antibody.

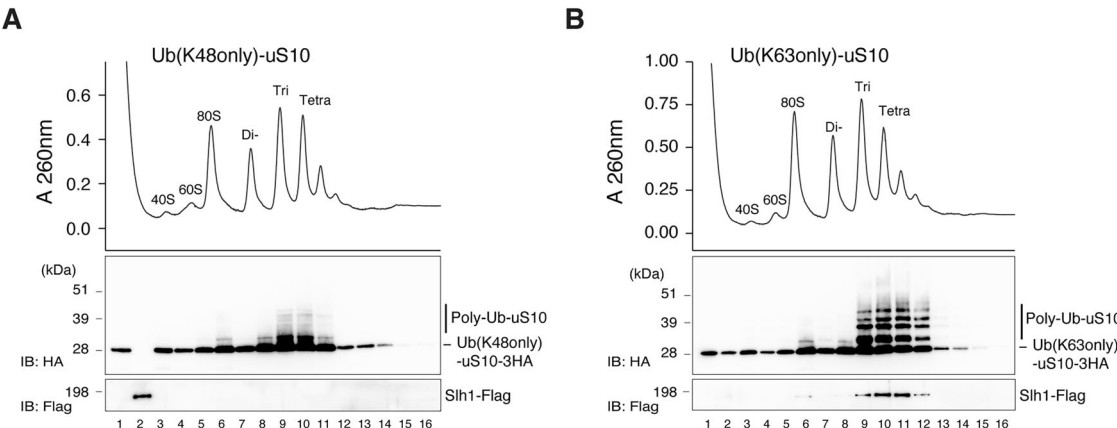

**Figure EV6. RQT complex is associated with the K63-linked ubiquitinated colliding ribosomes but not with K48-K63-mixed polyubiquitinated colliding ribosomes.**

The purified RNCs from the in vitro translation (IVT) reaction of the His-SDD1 model mRNA using the IVT extract prepared from Ub-K48only-uS10 mutant strain (**A**): *uS10Δski2Δ* expressing Ub-K48only-uS10-3HA from plasmid pST321, or Ub-K63only-uS10 mutant strain (**B**): *uS10Δski2Δ* expressing Ub-K63only-uS10-3HA from plasmid pST320, were incubated with the RQT complex in the absence of ATP and then separated by sucrose density gradient centrifugation. The ribosome abundance was detected by UV absorbance at 260 nm. HA-tagged uS10 and Flag-tagged Slh1, the component of the RQT complex, in each fraction, were detected by immunoblotting using anti-HA and anti-Flag antibodies, respectively.

