## [Peer Review File · The EMBO Journal]

Polyubiquitin architecture editing on collided ribosomes maintains persistent RQC activity

Shota Tomomatsu, Yoshitaka Matsuo, Fumiaki Ohtake, Takuya Tomita, Yasushi Saeki, and Toshifumi Inada

Corresponding author(s): Toshifumi Inada (toshiinada@ims.u-tokyo.ac.jp) , Yoshitaka Matsuo (yoshitaka-matsuo@g.ecc.u-tokyo.ac.jp)

Review Timeline:

Submission Date:	21st Apr 25
Editorial Decision:	2nd Jun 25
Revision Received:	25th Jul 25
Editorial Decision:	29th Aug 25
Revision Received:	1st Sep 25
Accepted:	4th Sep 25

Editor: Hartmut Vodermaier

Transaction Report:

Prof. Toshifumi Inada
The University of Tokyo
Division of RNA and gene regulation Institute of Medical Science,
4-6-1 shirokanedai
Minato-Ku, Tokyo 108-8639
Japan

2nd Jun 2025

Re: EMBOJ-2025-121138
Editing of the polyubiquitin architecture on the collided ribosome maintains persistent RQC activity

Dear Prof. Inada,

Thank you for submitting your study on polyubiquitin chain editing during ribosomal quality control to The EMBO Journal. I am sorry that it has taken somewhat longer than usually to obtain a complete set of referee reports and get back to you with a decision. We have now finally received the comments of three expert reviewers, which you will find copied below. Since all reviewers appreciate the overall technical quality as well as the potential interest of your results, we would be happy to consider this work further for publication, pending adequate revision in response to the referee reports.

As you will see, all referees bring up questions arising from the presented data, and feel that further understanding of certain aspects would be valuable to strengthen the significance of the study. In particular, it would be important to deepen the insights into the functional roles of each of the deubiquitinating enzymes and of the specific ubiquitin chain types - as queried especially by referee 2 (main concerns A, B, C), but also by referee 1 (point 3). I realize that not all of these points may warrant additional experimentation, but since we allow only a single (major) revision round, it will still be important to carefully respond to all queries at the time of resubmission. Therefore, I would encourage you to contact me with a revision plan and tentative point-by-point response to the full reviews already during the early stages of your revision work, so that we could discuss how key issues raised in the reports might best be resolved. We would also be open to extending the revision deadline if that should be helpful. Our 'scooping protection' (meaning that competing work appearing elsewhere in the meantime will not affect our considerations of your study) would of course remain valid also throughout such an extension.

Further information on preparing, formatting and uploading a revised manuscript can be found below and in our Guide to Authors; important presentational issues to improve would be language/writing (also with non-expert readers in mind) and updating the bibliography, making sure that all references are complete with year/volume/page numbers and in correct EMBO Journal format (alphabetical rather than numbered).

Thank you again for the opportunity to consider this work for The EMBO Journal, and I look forward to receiving your revised manuscript in due time.

Yours sincerely,

Hartmut Vodermaier

2) Each figure legend must specify
- size of the scale bars that are mandatory for all micrograph panels
- the statistical test used to generate error bars and P-values

- the type error bars (e.g., S.E.M., S.D.)
- the number (n) and nature (biological or technical replicate) of independent experiments underlying each data point
- Figures may not include error bars for experiments with $n < 3$; scatter plots showing individual data points should be used instead.

9) To facilitate reproducibility and cross-laboratory adoption of methodologies, please structure the Materials & Methods section as outlined in our guide to authors, including a completed Reagents and Tools Table that can be downloaded from our author guidelines as well (<https://www.embopress.org/page/journal/14602075/authorguide#structuredmethods>).

10) Digital image enhancement is acceptable practice, as long as it accurately represents the original data and conforms to community standards. If a figure has been subjected to significant electronic manipulation, this must be clearly noted in the figure legend and/or the 'Materials and Methods' section. The editors reserve the right to request original versions of figures and the original images that were used to assemble the figure. Finally, we generally encourage uploading of numerical as well as gel/blot image source data; for details see: embopress.org/page/journal/14602075/authorguide#sourcedata

In the interest of ensuring the conceptual advance provided by the work, we recommend submitting a revision within 3 months (31st Aug 2025). Please discuss the revision progress ahead of this time with the editor if you require more time to complete the revisions. Use the link below to submit your revision:

Link Not Available

Referee #1:

This manuscript by Tomomatsu S. et al. characterizes the role of the Ubp2 and Ubp3 deubiquitinases (DUBs) in Ribosome-associated Quality Control (RQC). The authors first demonstrate that deletion of either DUB impairs RQC function in yeast. Ubp2 is mildly associated with the 40S subunit, and its deletion leads to a marked increase in 40S and 80S ubiquitination. This is primarily driven by the addition of K63-linked chains on the uS10 subunit, suggesting an important role for Ubp2 in ribosome recycling. Ubp3 has a subtler effect on the polysome fraction, and its absence appears to cause accumulation of both K48 and K63 linkages on uS10, likely contributing to the maintenance of RQC efficiency during translation. Moreover, the authors show that the Hel2 ubiquitin ligase primarily extends K63 chains.

The authors present complementary *in vivo* and *in vitro* experiments and thoroughly examine the ubiquitin chain topology using ubiquitin mutants, chain-specific antibodies, and protein mass spectrometry. The experiments and their rationale are clearly introduced in a way that supports non-experts in RQC. The work is carefully executed, and the results are of high quality. There is some variability in the results concerning Ubp3 depending on the method used. For example, mass spectrometry shows an increase in K48 linkages in Figure 3B that is not apparent in the Western blot shown in Figure 3D. However, this likely reflects subtle differences not readily quantifiable by Western blotting, and potential variations introduced by the use of ubiquitin mutants or Hel2 overexpression. Nevertheless, the role of Ubp3 in RQC is convincingly demonstrated, and the authors' interpretations in the discussion are appropriate. Importantly, this study significantly advances our understanding of RQC and introduces several novel and valuable insights.

I recommend this high-quality manuscript for publication in The EMBO Journal, provided the authors address a series of relatively minor comments, as outlined below.

Specific Comments:

1. The authors should elaborate on why Ubp2 deletion causes a defect in RQC. For instance, impaired ribosome recycling may act as a "sink" that sequesters RQC components, thereby reducing their availability elsewhere in the cell.
2. If possible, quantitative analysis of the data shown in Figures 1A and 1B should be included to strengthen the initial evidence for the impact of DUB deletions on RQC.
3. If feasible, I suggest analyzing the K6/8R uS10 variant in *ubp2Δ* and *ubp3Δ* strains. While not essential given the breadth of the current dataset, such an experiment could help distinguish between poly-ubiquitination and multi-monoubiquitination, particularly in the context of Ubp3, which leads to the accumulation of lower-molecular-weight species. In this regard, it may also be more accurate to describe the changes as impacting "upper bands" or "high-molecular-weight species" rather than asserting poly-ubiquitylation (see page 6, line 8; similar care should be taken on page 8).
4. Please note somewhere that uS10 is also known as Rps20.
5. On page 8, line 1, replace "must disappear" with "are absent."
6. On line 9 of the same page, revise "the accumulation of all linkages type" to maybe "the accumulation of non-K63 linkages".
7. It should be noted that Ubp3's deubiquitinating activity on uS10 is approximately tenfold lower than that of Ubp2 (Figure 2E-F). The term "efficiently" on page 9, line 7, should be contextualized accordingly. Similarly, while Ubp3 appears to preferentially trim K48-linked chains, its overall impact is more modest compared to Ubp2 (see page 10, line 2).
8. It is unclear why the potential migration of K29/K63 bands is indicated in Figures 5D and 5E, as no bands are visible in the upper portion of the gels. Clarification is recommended.
9. On page 15, line 6, avoid using the term "significantly" unless the difference was statistically assessed.
10. I was curious why reagents such as NEM or iodoacetamide were not used during cell lysis to inhibit potential post-lysis DUB activity.

Finally, I apologize for the delay in submitting this review (faster turnaround was simply not possible this time) and thank the authors for their patience.

Referee #2:

In this manuscript, Tomomatsu et al. identify Ubp2 and Ubp3 in *S. cerevisiae* as de-ubiquitinating enzymes that shorten ubiquitin (Ub)-chains on the ribosomal protein uS10 upon ribosome stalling. The authors further propose that both Ubp2 and Ubp3 promote ribosome-associated quality control (RQC), i.e. the pathway that triggers dissociation of stalled ribosomes and degradation of the corresponding nascent polypeptide. This observation is rather surprising as poly-ubiquitination of uS10 is known to be a key step in initiating RQC, and one would expect that shortening of these Ub chains by Ubp2/3 should antagonize RQC.

The authors further show that Ubp2 specifically de-ubiquitinates K63-linked Ub chains on uS10 associated with free 40S subunits, whereas Ubp3 cleaves all Ub chains on uS10 associated with translating ribosomes. Using carefully designed approaches including chimeric Ub-uS10 fusion proteins, the authors then find that Ub chains on uS10 have mixed K29/K48/K63-linkages, and identify the E3-ligase Ufd4 as being required for K29-linked Ub on uS10. Making use of an elegant functional *in vitro* assay, the authors demonstrate that only K63-linked Ub chains on uS10 trigger RQC, while K48-linked Ub chains fail to do so.

This manuscript offers a wealth of findings related to RQC, and from a technical point of view it fulfills high standards with a range of innovative approaches addressing the nature of mixed Ub-chains on uS10 during ribosome stalling, and identifying enzymes involved in this process. In some cases, better quantification would be helpful. My main concern is that the manuscript does not solve the functional role of the de-ubiquitinating enzymes and/or the relevance of the mixed Ub chains: A) If Ubp3 can de-ubiquitinate both K48- and K63-linked chains on uS10 associated with polysomes, why does it not antagonize RQC by generally shortening the chains? B) Since Ubp2 is associated with free 40S subunits, how is it possible that it affects RQC at all? It seems more likely that Ubp2 is involved in deciding whether ubiquitinated 40S subunits after dissociation are recycled or degraded, yet this possibility is not explored by the authors. C) While the authors clearly show that K48-linked Ub does not induce RQC *in vitro*, it remains unclear whether K48 linkages in cells indeed serve to suppress RQC. Experiments addressing the functional role

of the identified enzymes and mixed Ub linkages would strengthen the manuscript.

Specific comments:

Fig.1A and EV1B: Size of uS10-3HA shift from about 20 kDa to about 32 kDa is more than what is expected for a single Ubiquitin molecule. This contrasts with the shift from 14 to 19 kDa the same authors observed previously when looking at mono-Ub of endogenous (untagged) uS10 (PMID: 36302773) or HA-tagged uS10 (PMID: 36627279). The shift seen with uS10-3HA in the present manuscript appears too large for a single Ubiquitin, also when comparing to the shift to next bands corresponding to di-Ub- and tri-Ub-uS10-3HA. Could the first modification be a higher MW member of the ubiquitin-like protein family? To assess this further, the authors may consider expressing tagged ubiquitin. Western blot analysis against the tag would then indicate whether the first modification on uS10-3HA is indeed Ub.

Quantification of effects: In some cases, the authors show a single Western blot to document their results (e.g. changes in uS10-Ub levels in Fig.1A, changes in arrest products in Fig.1B). Besides showing exemplary Western blots, the authors need to quantify the bands in question from a reasonable number of biological repeat experiments, and show results in graphs depicting both the individual measurements and the mean values {plus minus} SD. This will allow them to test if the observed changes are statistically significant, and substantiate rather vague descriptions in the text (e.g. "...arrest products were gently reduced in ... mutants", page 6). This is of particular importance for the ubp3 deletion, where the increase Ub-uS10 is sometimes weak (Fig.1A) and sometimes not visible (Fig.4F).

The author show that Ubp2 and Ubp3 (slightly) enhance RQC, yet is not clear whether this activity occurs via de-ubiquitination of Ub-uS10, or possibly other targets. Can the authors exclude the latter possibility?

Fig.EV2B: Interestingly Ubp3 migrates exclusively with heavy polysomes. The authors should disassemble polysomes, e.g. by RNase treatment of the lysate, do test if Ubp3 is indeed associated with heavy polysomes, or possibly with another high MW complex in the cell. If the authors' interpretation can be confirmed, the results in Fig.EV2A and B should go into the main figures as they suggest a fundamental difference between the functions of Ubp2 and Ubp3.

Fig.2C: Since the authors show absolute intensities, it is unclear how the Western blot signals were normalized between different repeat experiments. Signal intensities typically vary between biological repeats depending on exposure time etc. - some internal normalization is needed.

If Ubp2 trims Ub chains on uS10 of free 40S subunits after RQC-induced subunit dissociation, it is not clear how Ubp2 can have an effect on CAT-tailing of the nascent polypeptide as shown in Fig.1.

In Fig.6F, it is not clear how the different tetra-Ub chains were expressed or generated.

Fig.7A: The description in the text (page 15) is strange: "As expected, the colliding ribosomes conjugated with a K48/K63-mixed-linkage ubiquitin chain were not affected after the splitting reaction (Fig. 7A)." There is no dissociation of ribosomes (splitting reaction) with the K48-linked Ub, so "after the splitting reaction" does not make sense.

In the abstract and at the beginning of the results section, the authors should state that their study was conducted in *Saccharomyces cerevisiae*.

page 12: The sentence "Ub-uS10 with K63-linked mono-ubiquitin was detected in Ub-uS10 wildtype but not in all mutants containing Ub-K63R mutation (Fig. 5D, lanes 1, 4, 6-8)." is strange, it should say "...was detected in ... wildtype but in none of the mutants containing Ub-K63R mutation (Fig. 5D, lanes 1, 4, 6-8)."

The manuscript would benefit from editing of the English language, especially with regard to the use of articles.

Referee #3:

In this manuscript, Tomomatsu et al., investigate the nature of the ubiquitin chains that build on the uS10 ribosomal subunit in the context of colliding ribosome conditions. They identify 2 de-ubiquitinating enzymes that edit the ubiquitinated uS10, Ubp2 and Ubp3, and find that they play different roles. Ubp2 appears to remove K63-linked polyubiquitin chains and this on ribosomes after splitting, whereas Ubp3 removes chains from translating ribosomes and can cleave both 48-linked and 63-linked chains. They also show that there are both 63-linkages and 48-linkages on uS10, and that the 48-linkage is negatively impacting the RQC-mediated response. Taken together, their findings indicate that the 2 DUBs contribute in different ways to regulate the RQC response.

General

This manuscript reports a very solid study with well-controlled experiments that show clear results that are also well presented. It provides important clarification on the regulation of ribosome ubiquitination during translation elongation now understood to play a key role in cellular responses to stress. This is a topic that has been attracting more and more attention in recent years and is of general interest.

I do not have a lot of comments, because overall the study is very convincing, but a few specific comments outlined below:

- 1) On the bottom of the first page of the introduction there is an issue with fonts and font size (references 26-30).
- 2) On figure 1A there seems to be an impact on overall uS10 protein levels in the mutants, it would be nice to have a loading control
- 3) On figure 1C, the impact of *ubc2Δ* is very minor compared to that of *ubc3Δ* it would be relevant to mention this in the text
- 4) On the first page of the results section, right after (Fig.1B), I would replace "suggesting" with "correlating", because at this point, "suggesting" is too strong.
- 5) When Hel2 is introduced, at the end of the first section of the results it mentions: "...is required to produce arrest products...."; I feel that this is somewhat an short-cut statement, and for readers who do not know the field, does not really explain the role of Hel2. It could be explained better
- 6) For figure 2C, is it overall ubiquitination that is quantified, or for each mutant the specific ubiquitination that changes as mentioned in the text? This needs clarification, because if it is overall ubiquitination, then the mention of the specific forms affected by each mutant is evaluated by eye only.
- 7) The authors mention that they cannot detect ubiquitinated unassembled uS10, which may be degraded by the proteasome. Could it not be that since it is additional uS10 expressed from a plasmid, it may just not get ubiquitinated if it did not integrate ribosomes?
- 8) It is quite striking the difference in concentrations of Ubp2 and Ubp3 needed to see an effect in the in vitro assay. Can the authors comment? Is this related to the quality of the enzyme preparations? Is the concentration of the 2 enzymes in cells different?
- 9) The authors use Hel2 overexpression to see more ubiquitination of uS10. Does this mean that Hel2 is limiting in cells and that not all collisions result in ubiquitination?

Point by point response to Reviewers (Submission ID: EMBOJ-2025-121138R)

We thank all reviewers for their positive, helpful, and insightful comments. In our detailed response, the reviewers' comments are *Italicized* whereas our response is in Roman typeface with blue color.

Referee #1:

This manuscript by Tomomatsu S. et al. characterizes the role of the Ubp2 and Ubp3 deubiquitinases (DUBs) in Ribosome-associated Quality Control (RQC). The authors first demonstrate that deletion of either DUB impairs RQC function in yeast. Ubp2 is mildly associated with the 40S subunit, and its deletion leads to a marked increase in 40S and 80S ubiquitination. This is primarily driven by the addition of K63-linked chains on the uS10 subunit, suggesting an important role for Ubp2 in ribosome recycling. Ubp3 has a subtler effect on the polysome fraction, and its absence appears to cause accumulation of both K48 and K63 linkages on uS10, likely contributing to the maintenance of RQC efficiency during translation. Moreover, the authors show that the Hel2 ubiquitin ligase primarily extends K63 chains.

The authors present complementary in vivo and in vitro experiments and thoroughly examine the ubiquitin chain topology using ubiquitin mutants, chain-specific antibodies, and protein mass spectrometry. The experiments and their rationale are clearly introduced in a way that supports non-experts in RQC. The work is carefully executed, and the results are of high quality. There is some variability in the results concerning Ubp3 depending on the method used. For example, mass spectrometry shows an increase in K48 linkages in Figure 3B that is not apparent in the Western blot shown in Figure 3D. However, this likely reflects subtle differences not readily quantifiable by Western blotting, and potential variations introduced by the use of ubiquitin mutants or Hel2 overexpression. Nevertheless, the role of Ubp3 in RQC is convincingly demonstrated, and the authors' interpretations in the discussion are appropriate. Importantly, this study significantly advances our understanding of RQC and introduces several novel and valuable insights.

I recommend this high-quality manuscript for publication in The EMBO Journal, provided the authors address a series of relatively minor comments, as outlined below.

Specific Comments:

1. The authors should elaborate on why *Ubp2* deletion causes a defect in RQC. For instance, impaired ribosome recycling may act as a "sink" that sequesters RQC components, thereby reducing their availability elsewhere in the cell.

We agree with the reviewer's comment. It is well established that ubiquitinated collided ribosomes are disassembled by the RQT complex (Matsuo *et al.*, 2020 *Nat Struct Mol Biol* & Matsuo *et al.*, 2023 *Nat Commun*), and cryo-EM has revealed that the RQT complex remains bound to the 40S subunit after subunit dissociation (Best *et al.*, 2023 *Nat Commun*). Cue3 and Rqt4, components of the RQT complex, interact with K63-linked ubiquitin chains attached to the collided ribosomes, thereby recruiting the RNA helicase Slh1 to the stalled ribosomes (Matsuo *et al.*, 2023 *Nat Commun*). Therefore, it is plausible that the RQT complex remains tethered to the 40S subunit via the ubiquitin chain even after subunit dissociation. This suggests that *Ubp2*-mediated removal of K63-linked ubiquitin chains from the 40S subunit may facilitate recycling of the RQT complex. Consequently, loss of *Ubp2* could reduce the pool of available RQT complexes, thereby impairing RQC activity. We have explicitly incorporated this discussion into the revised manuscript (page 16, lanes 436-446).

2. If possible, quantitative analysis of the data shown in Figures 1A and 1B should be included to strengthen the initial evidence for the impact of DUB deletions on RQC.

Following the reviewer's suggestion, we added quantification results (N = 3) for the Western blotting shown in revised Figure 1A and Figure 1B as follows.

3. If feasible, I suggest analyzing the K6/8R uS10 variant in *ubp2Δ* and *ubp3Δ* strains. While not essential given the breadth of the current dataset, such an experiment could help distinguish between poly-ubiquitination and multi-monoubiquitination, particularly in the context of Ubp3, which leads to the accumulation of lower-molecular-weight species. In this regard, it may also be more accurate to describe the changes as impacting "upper bands" or "high-molecular-weight species" rather than asserting poly-ubiquitylation (see page 6, line 8; similar care should be taken on page 8).

In response to the reviewer's suggestion, we analyzed the ubiquitination of the uS10-K6/8R mutant in the *ubp2Δ* and *ubp3Δ* strains (revised Figure EV1C; page6, lanes 118-120). All upper bands corresponding to ubiquitinated uS10 disappeared in the uS10-K6/8R mutant, even in the absence of Ubp2 or Ubp3, indicating that these bands are formed through ubiquitination at Lys6 and/or Lys8. However, we cannot completely exclude the possibility that the upper bands also contain di-monoubiquitinated species at Lys6 and Lys8 in addition to polyubiquitinated forms. Therefore, we have revised the wording from "polyubiquitination of uS10" to "high-molecular-weight of ubiquitinated uS10" (page6, lane 130).

4. Please note somewhere that uS10 is also known as Rps20.

We have indicated in the Introduction that uS10 corresponds to Rps20 upon its first mention. (Page 3, lane 54)

5. On page 8, line 1, replace "must disappear" with "are absent."

We have replaced "must disappear" with "are absent", as suggested by the reviewer (Page 8, lane 190).

6. On line 9 of the same page, revise "the accumulation of all linkages type" to maybe "the accumulation of non-K63 linkages".

We have revised the phrase from "the accumulation of all linkages type" to "the accumulation of non-K63 linkages," as suggested (Page 8, lane 198-199).

7. It should be noted that Ubp3's deubiquitinating activity on uS10 is approximately tenfold lower than that of Ubp2 (Figure 2E-F). The term "efficiently" on page 9, line 7, should be contextualized accordingly. Similarly, while Ubp3 appears to preferentially trim K48-linked chains, its overall impact is more modest compared to Ubp2 (see page 10, line 2).

Thank you for the very insightful comment. We have revised the sentence "In contrast to Ubp2, Ubp3-WT efficiently cleaved the ubiquitin chain on uS10-K6R across all the combinations of genomic backgrounds (Fig. 2F)" as follows:

"Although Ubp3 exhibits lower DUB activity than Ubp2 and requires approximately a tenfold higher concentration to remove ubiquitin chains in vitro, it nonetheless cleaved polyubiquitin chains on uS10-K6R across all combinations of substrate types and genomic backgrounds (Fig. 2F)." (Page 9, Lanes 228-230)

8. It is unclear why the potential migration of K29/K63 bands is indicated in Figures 5D and 5E, as no bands are visible in the upper portion of the gels. Clarification is recommended.

We apologize for the confusion. Due to limited space at the appropriate arrow positions, the labels above were intended to indicate that the corresponding arrows refer to K29/K63 or K63 linkages (Figure 5D-F), and were not meant to point to specific bands. To avoid any misunderstanding, we have revised the figure accordingly.

9. On page 15, line 6, avoid using the term "significantly" unless the difference was statistically assessed.

We appreciate the reviewer's point and will revise the wording to avoid using "significantly," since the difference was not statistically tested. (Page 15, line 413)

10. I was curious why reagents such as NEM or iodoacetamide were not used during cell lysis to inhibit potential post-lysis DUB activity.

We apologize for the oversight. This was our mistake. For Western blotting, we lyse the cells by adding TCA and disrupting them with zirconia beads. Since TCA inhibits DUB activity, we do not add any DUB inhibitors during this procedure. However, for the experiments involving ribosome purification and observation of ubiquitin chains, we do include PR-619 as a DUB inhibitor. It appears that this detail was missing from the Methods section, and we have now explicitly included this point in the revised manuscript. (Page 21, line 577 & Page 22, line 612)

Finally, I apologize for the delay in submitting this review (faster turnaround was simply not possible this time) and thank the authors for their patience.

Referee #2:

In this manuscript, Tomomatsu et al. identify Ubp2 and Ubp3 in S. cerevisiae as de-ubiquitinating enzymes that shorten ubiquitin (Ub)-chains on the ribosomal protein uS10 upon ribosome stalling. The authors further propose that both Ubp2 and Ubp3 promote ribosome-associated quality control (RQC), i.e. the pathway that triggers dissociation of stalled ribosomes and degradation of the corresponding nascent polypeptide. This observation is rather surprising as poly-ubiquitination of uS10 is known to be a key step in initiating RQC, and one would expect that shortening of these Ub chains by Ubp2/3 should antagonize RQC.

The authors further show that Ubp2 specifically de-ubiquitinates K63-linked Ub chains on uS10 associated with free 40S subunits, whereas Ubp3 cleaves all Ub chains on uS10 associated with translating ribosomes. Using carefully designed approaches including chimeric Ub-uS10 fusion proteins, the authors then find that Ub chains on uS10 have mixed K29/K48/K63-linkages, and identify the E3-ligase Ufd4 as being required for K29-linked Ub

on uS10. Making use of an elegant functional *in vitro* assay, the authors demonstrate that only K63-linked Ub chains on uS10 trigger RQC, while K48-linked Ub chains fail to do so.

This manuscript offers a wealth of findings related to RQC, and from a technical point of view it fulfils high standards with a range of innovative approaches addressing the nature of mixed Ub-chains on uS10 during ribosome stalling, and identifying enzymes involved in this process. In some cases, better quantification would be helpful.

My main concern is that the manuscript does not solve the functional role of the de-ubiquitinating enzymes and/or the relevance of the mixed Ub chains:

A) If Ubp3 can de-ubiquitinate both K48- and K63-linked chains on uS10 associated with polysomes, why does it not antagonize RQC by generally shortening the chains?

Although this point is briefly mentioned in the Discussion, we hypothesize that K63-linked ubiquitin chains on collided ribosomes are rapidly recognized by the RQT complex, leading to subunit dissociation. *In vitro* DUB assays (Figure 2E & 2F) have shown that the enzymatic activity of Ubp3 is substantially lower than that of Ubp2. Due to this weak activity, it is possible that Ubp3 cannot effectively compete with the RQT complex for binding to K63-linked chains on collided ribosomes. In contrast, when K48-linked chains are present, the RQT complex is unable to bind, potentially allowing Ubp3 to remove these chains. In the revised manuscript, we have emphasized this competitive relationship between the RQT complex and Ubp3 to better explain why Ubp3 does not inhibit RQC by trimming K63-linked ubiquitin chains. (Page 17, lanes 474-477)

B) Since Ubp2 is associated with free 40S subunits, how is it possible that it affects RQC at all? It seems more likely that Ubp2 is involved in deciding whether ubiquitinated 40S subunits after dissociation are recycled or degraded, yet this possibility is not explored by the authors.

Thank you for the very insightful comment. This point overlaps with the first comment raised by Reviewer #1.

It is well established that ubiquitinated collided ribosomes are disassembled by the RQT complex (Matsuo *et al.*, 2020 *Nat Struct Mol Biol* & Matsuo *et al.*, 2023 *Nat Commun*), and cryo-EM has revealed that the RQT complex remains bound to the 40S subunit after subunit dissociation (Best *et al.*, 2023 *Nat Commun*). Cue3 and Rqt4, components

of the RQT complex, interact with K63-linked ubiquitin chains attached to the collided ribosomes, thereby recruiting the RNA helicase Slh1 to the stalled ribosomes (Matsuo *et al.*, 2023 *Nat Commun*). Therefore, it is plausible that the RQT complex remains tethered to the 40S subunit via the ubiquitin chain even after subunit dissociation. This suggests that Ubp2-mediated removal of K63-linked ubiquitin chains from the 40S subunit may facilitate recycling of the RQT complex. Consequently, loss of Ubp2 could reduce the pool of available RQT complexes, thereby impairing RQC activity. We have explicitly incorporated this discussion into the revised manuscript. (Page 16, lanes 436-446)

We appreciate the reviewer's insightful comment. However, it should be noted that the ubiquitin chains implicated in the degradation of the 40S ribosomal subunit through the 18S non-functional rRNA decay (18S NRD) pathway are conjugated to the ribosomal protein uS3, rather than uS10 (Sugiyama *et al.*, 2019 *Cell Rep*). Given that uS10-linked ubiquitin chains are not involved in this specific degradation process, we believe that the 18S NRD pathway is not directly relevant to the mechanisms examined in the present study. Nonetheless, we cannot rule out the possibility that Ubp2 targets uS3 ubiquitination and is also involved in rRNA degradation.

C) While the authors clearly show that K48-linked Ub does not induce RQC in vitro, it remains unclear whether K48 linkages in cells indeed serve to suppress RQC. Experiments addressing the functional role of the identified enzymes and mixed Ub linkages would strengthen the manuscript.

We agree with the reviewer's comment. However, demonstrating the inhibitory role of K48-linked ubiquitin chains in RQC *in vivo* is technically quite challenging. Therefore, in this study, we investigated this possibility using an *in vitro* system. We have stated in the discussion section that future studies are needed to determine whether K48 linkages suppress RQC *in vivo*. (Page 18, lanes 498-499)

Specific comments:

Fig.1A and EV1B: Size of uS10-3HA shift from about 20 kDa to about 32 kDa is more than what is expected for a single Ubiquitin molecule. This contrasts with the shift from 14 to 19 kDa the same authors observed previously when looking at mono-Ub of endogenous (untagged) uS10 (PMID: 36302773) or HA-tagged uS10 (PMID: 36627279). The shift seen with uS10-3HA in the present manuscript appears too large for a single Ubiquitin, also when

comparing to the shift to next bands corresponding to di-Ub- and tri-Ub-uS10-3HA. Could the first modification be a higher MW member of the ubiquitin-like protein family?

To assess this further, the authors may consider expressing tagged ubiquitin. Western blot analysis against the tag would then indicate whether the first modification on uS10-3HA is indeed Ub.

We apologize for the confusion. This phenomenon is primarily due to differences in electrophoretic migration patterns caused by variations in gel composition and running buffer conditions. Although the study with PMID: 36302773 analyzed mammalian uS10 and is therefore not directly comparable, we have presented results obtained under the electrophoresis conditions (10% NuPAGE & MOPS buffer) used in the study with PMID: 36627279 to demonstrate that there is no discrepancy (see the red arrow below).

As suggested by the reviewer, expressing tagged ubiquitin and performing Western blotting with an antibody against the tag would detect multiple ubiquitinated proteins in addition to uS10, making it difficult to specifically identify uS10 ubiquitination.

Quantification of effects: In some cases, the authors show a single Western blot to document their results (e.g. changes in uS10-Ub levels in Fig.1A, changes in arrest products in Fig.1B). Besides showing exemplary Western blots, the authors need to quantify the bands in question from a reasonable number of biological repeat experiments, and show results in graphs depicting both the individual measurements and the mean values {plus minus} SD. This will allow them to test if the observed changes are statistically significant, and substantiate rather vague des articular importance for the ubp3 deletion, where the increase Ub-uS10 is sometimes weak (Fig.1A) and sometimes not visible (Fig.4F).

This point overlaps with the second comment raised by Reviewer #1.

Following the reviewer's suggestion, we have included quantification results (N = 3) for the Western blotting shown in the revised Figure 1A and Figure 1B.

The author show that *Ubp2* and *Ubp3* (slightly) enhance RQC, yet is not clear whether this activity occurs via de-ubiquitination of Ub-uS10, or possibly other targets. Can the authors exclude the latter possibility?

As the reviewer correctly noted, we could not entirely exclude the latter possibility. We had stated the additional function of *Ubp3* in the Discussion section. These functions might be indirectly involved in RQC activity.

Fig.EV2B: Interestingly *Ubp3* migrates exclusively with heavy polysomes. The authors should disassemble polysomes, e.g. by RNase treatment of the lysate, do test if *Ubp3* is indeed associated with heavy polysomes, or possibly with another high MW complex in the cell. If the authors' interpretation can be confirmed, the results in *Fig.EV2A* and *B* should go into the main figures as they suggest a fundamental difference between the functions of *Ubp2* and *Ubp3*.

To address the reviewer's request, we examined whether treatment with micrococcal

nuclease causes Ubp3 to shift to lighter fractions. As expected, Ubp3 predominantly shifted from the polysome fractions to the disome fraction after digestion by MNase (Revised Figure EV2B, Page 7, lane 171; see below).

Fig.2C: Since the authors show absolute intensities, it is unclear how the Western blot signals were normalized between different repeat experiments. Signal intensities typically vary between biological repeats depending on exposure time etc. - some internal normalization is needed.

Normalization was performed using the unmodified ribosomal protein uS10. This information has now been clearly stated in the revised figure legend. (Pages 34, lanes 1015-1017)

If Ubp2 trims Ub chains on uS10 of free 40S subunits after RQC-induced subunit dissociation, it is not clear how Ubp2 can have an effect on CAT-tailing of the nascent polypeptide as shown in Fig.1.

As described above, we suggest that the loss of Ubp2 inhibits recycling of the RQT complex, thereby reducing the efficiency of subunit dissociation of stalled ribosomes. Consequently, a decrease in the amount of arrest products is expected to be accompanied by a reduction in CAT-tailing.

In Fig.6F, it is not clear how the different tetra-Ub chains were expressed or generated.

We have clarified that the different tetra-ubiquitin chains used in Fig. 6F were commercially purchased from UBPBio or R&D systems. This point has been stated in the revised manuscript. (Page 24, lane 679)

Fig.7A: The description in the text (page 15) is strange: "As expected, the colliding ribosomes conjugated with a K48/K63-mixed-linkage ubiquitin chain were not affected after the splitting reaction (Fig. 7A)." There is no dissociation of ribosomes (splitting reaction) with the K48-linked Ub, so "after the splitting reaction" does not make sense.

We appreciate the reviewer's comment. We have revised the sentence "As expected, the colliding ribosomes conjugated with a K48/K63-mixed-linkage ubiquitin chain were not affected after the splitting reaction (Fig. 7A).", as follows:

"As expected, the colliding ribosomes conjugated with a K48/K63-mixed-linkage ubiquitin chain were not affected after the addition of the RQT complex in the presence of ATP (Fig. 7A)." (Page 15, lanes 411-412)

*In the abstract and at the beginning of the results section, the authors should state that their study was conducted in *Saccharomyces cerevisiae*.*

We have clearly stated in the beginning of the results section that the study was conducted in *Saccharomyces cerevisiae*. (Page 6, Lane 124)

page 12: The sentence "Ub-uS10 with K63-linked mono-ubiquitin was detected in Ub-uS10 wildtype but not in all mutants containing Ub-K63R mutation (Fig. 5D, lanes 1, 4, 6-8)." is strange, it should say "...was detected in ... wildtype but in none of the mutants containing Ub-K63R mutation (Fig. 5D, lanes 1, 4, 6-8)."

Thank you for pointing this out. We have revised the sentence as suggested to:

"Ub-uS10 with K63-linked mono-ubiquitin was detected in Ub-uS10 wildtype but in none of the mutants containing the Ub-K63R mutation (Fig. 5D, lanes 1, 4, 6-8)." (Page 13, lanes 342-343)

The manuscript would benefit from editing of the English language, especially with regard to the use of articles.

We appreciate the reviewer's valuable feedback. In response, we have carefully revised the manuscript to improve the English, with particular attention to the use of articles.

Referee #3:

In this manuscript, Tomomatsu et al., investigate the nature of the ubiquitin chains that build on the uS10 ribosomal subunit in the context of colliding ribosome conditions. They identify 2 de-ubiquitinating enzymes that edit the ubiquitinated uS10, Ubp2 and Ubp3, and find that they play different roles. Ubp2 appears to remove K63-linked polyubiquitin chains and this on ribosomes after splitting, whereas Ubp3 removes chains from translating ribosomes and can cleave both 48-linked and 63-linked chains. They also show that there are both 63-linkages and 48-linkages on uS10, and that the 48-linkage is negatively impacting the RQC-mediated response. Taken together, their findings indicate that the 2 DUBs contribute in different ways to regulate the RQC response.

General

This manuscript reports a very solid study with well-controlled experiments that show clear results that are also well presented. It provides important clarification on the regulation of ribosome ubiquitination during translation elongation now understood to play a key role in cellular responses to stress. This is a topic that has been attracting more and more attention in recent years and is of general interest.

I do not have a lot of comments, because overall the study is very convincing, but a few specific comments outlined below:

1) On the bottom of the first page of the introduction there is an issue with fonts and font size (references 26-30).

Thank you for pointing this out. We have corrected the font and font size issues for references 26–30 at the bottom of the first page of the Introduction.

2) On figure 1A there seems to be an impact on overall uS10 protein levels in the mutants, it would be nice to have a loading control

We appreciate the reviewer's comment. We have included a loading control in revised Figure 1A to confirm equal protein loading.

3) *On figure 1C, the impact of ubc2Δ is very minor compared to that of ubc3Δ it would be relevant to mention this in the text*

We had thought we had mentioned in the original text that the impact of *ubp2Δ* is minor compared to that of *ubp3Δ* in Figure 1C as follows (Page7, lanes 154-159):

The observation of growth rates in the presence of low-dose anisomycin clearly showed mild and severe growth defects in *ubp2Δ* and *ubp3Δ* single deletion mutants, respectively, and the double deletion of these DUBs displayed a synergistic effect (**Fig. 1C**). Since Ubp3 is involved in several mechanisms, including ribophagy and proteasome degradation pathway, the *ubp3Δ* deletion mutant could display a higher impact on the growth defects due to low-dose anisomycin treatment than the *ubp2Δ* deletion mutant.

4) *On the first page of the results section, right after (Fig. 1B), I would replace "suggesting" with "correlating", because at this point, "suggesting" is too strong.*

We have replaced "suggesting" with "correlating" after (Fig. 1B) on the first page of the results section as recommended. (Page 6, lane 147)

5) *When Hel2 is introduced, at the end of the first section of the results it mentions: "...is required to produce arrest products...."; I feel that this is somewhat an short-cut statement, and for readers who do not know the field, does not really explain the role of Hel2. It could be explained better*

We appreciate the reviewer's insightful comment. In response to the suggestion, we have included a clearer introduction to the role of Hel2 to aid understanding for readers who are less familiar with the field. (Page 7, lanes 160-161)

6) *For figure 2C, is it overall ubiquitination that is quantified, or for each mutant the specific ubiquitination that changes as mentioned in the text? This needs clarification, because if it is overall ubiquitination, then the mention of the specific forms affected by each mutant is evaluated by eye only.*

We thank the reviewer for the helpful comment. To clarify, the quantification in Figure 2C reflects overall ubiquitination levels. We have revised the text in the figure legends to make this point clearly. (Pages 35, lanes 1018-1021)

7) *The authors mention that they cannot detect ubiquitinated unassembled uS10, which may be degraded by the proteasome. Could it not be that since it is additional uS10 expressed from a plasmid, it may just not get ubiquitinated if it did not integrate ribosomes?*

We fully agree with the reviewer's insightful comment. As suggested, it is indeed possible that the plasmid-expressed uS10 may not be ubiquitinated simply because it fails to be incorporated into ribosomes. We have included this possibility in the revised manuscript. (Page 7-8, lanes 180-182)

8) *It is quite striking the difference in concentrations of Ubp2 and Ubp3 needed to see an effect in the in vitro assay. Can the authors comment? Is this related to the quality of the enzyme preparations? Is the concentration of the 2 enzymes in cells different?*

Although the in vitro enzymatic activity of Ubp3 is approximately 10-fold lower than that of Ubp2, the estimated intracellular copy numbers of Ubp3 and Ubp2 are about 6,000 and 2,000, respectively (Chong *et al.*, 2015 *Cell*), indicating only a ~3-fold difference. As briefly mentioned above, we speculate on the difference in enzymatic activity between Ubp2 and Ubp3 based on their catalytic properties and substrate specificities. Ubp2 likely requires high activity to rapidly remove K63-linked ubiquitin chains from the 40S subunit after ribosomal subunit dissociation. In contrast, although Ubp3 is capable of removing both K63- and K48-linked chains in vitro, it does not interfere with RQC activity. This suggests that the K63-linked chains on collided ribosomes need to be promptly recognized by the RQT complex, and therefore, the lower activity of Ubp3 may be an inherent feature that prevents it from prematurely removing these chains. We have included this discussion in the revised manuscript. (Page 17, lanes 474-477)

9) *The authors use Hel2 overexpression to see more ubiquitination of uS10. Does this mean that Hel2 is limiting in cells and that not all collisions result in ubiquitination?*

The likelihood of this scenario depends on the frequency of ribosome collisions in cells, but considering that the expression level of Hel2 is not particularly high relative to the abundance of ribosomes, and that overexpression of Hel2 leads to increased ubiquitination, it is highly likely that not all collisions result in ubiquitination, as the reviewer suggested.

Prof. Toshifumi Inada
The University of Tokyo
Division of RNA and gene regulation Institute of Medical Science,
4-6-1 shirokanedai
Minato-Ku, Tokyo 108-8639
Japan

29th Aug 2025

Re: EMBOJ-2025-121138R
Editing of the polyubiquitin architecture on the collided ribosome maintains persistent RQC activity

Dear Toshifumi ,

Thank you for submitting your revised manuscript to The EMBO Journal. Two of the original referees have now assessed it once more, and were generally satisfied with the revisions. Referee 3 still notes a few presentational/textual issues that would need to be incorporated during a final round of minor revision. In addition, please also address the following remaining editorial issues at this stage:

- Please carefully go through the reference list and make sure that each reference is complete with citation year, volume, and page/locator numbers - this information is currently missing for several of them.
- When referring to the Appendix tables in the text, please make sure to maintain the correct naming ("Appendix Table S1/2").
- Please check once more through the "Methods" section - e.g. in the first section, there is no reference(s) for "as previously described". Furthermore, in several instances μL or μM are wrongly written with a "u" instead of the Greek "Mu" symbol. And in places where you should have reused passages from previous papers word-by-word, please make sure to directly refer to the respective articles appropriately.
- Finally, please provide suggestions for a short 'blurb' text prefacing and summing up the study in two sentences (max. 250 characters), followed by 3-5 one-sentence 'bullet points' with brief factual statements of key results of the paper; they will form the basis of an editor-written 'Synopsis' accompanying the online version of the article. Please also upload a synopsis image, which can be used as a "visual title" for the synopsis section of your paper. The image should be in PNG or JPG format with the modest dimensions of EXACTLY 550 pixels wide and between 300 and 600 pixels high.

I am returning the manuscript to you for a final round of minor revision, solely to allow you to make these modifications and upload the revised files. Once we will have received them, we should be ready to swiftly proceed with formal acceptance and production of the manuscript.

With kind regards,

Hartmut

- 1) Every manuscript requires a Data Availability section (even if only stating that no deposited datasets are included). Primary datasets or computer code produced in the current study have to be deposited in appropriate public repositories prior to resubmission, and reviewer access details provided in case that public access is not yet allowed. Further information: embopress.org/page/journal/14602075/authorguide#dataavailability
- 2) Each figure legend must specify
 - size of the scale bars that are mandatory for all micrograph panels
 - the statistical test used to generate error bars and P-values

- the type error bars (e.g., S.E.M., S.D.)
- the number (n) and nature (biological or technical replicate) of independent experiments underlying each data point
- Figures may not include error bars for experiments with $n < 3$; scatter plots showing individual data points should be used instead.

9) To facilitate reproducibility and cross-laboratory adoption of methodologies, please structure the Materials & Methods section as outlined in our guide to authors, including a completed Reagents and Tools Table that can be downloaded from our author guidelines as well (<https://www.embopress.org/page/journal/14602075/authorguide#structuredmethods>).

10) Digital image enhancement is acceptable practice, as long as it accurately represents the original data and conforms to community standards. If a figure has been subjected to significant electronic manipulation, this must be clearly noted in the figure legend and/or the 'Materials and Methods' section. The editors reserve the right to request original versions of figures and the original images that were used to assemble the figure. Finally, we generally encourage uploading of numerical as well as gel/blot image source data; for details see: embopress.org/page/journal/14602075/authorguide#sourcedata

In the interest of ensuring the conceptual advance provided by the work, we recommend submitting a revision within 3 months (27th Nov 2025). Please discuss the revision progress ahead of this time with the editor if you require more time to complete the revisions. Use the link below to submit your revision:

Link Not Available

Referee #1:

The authors have adequately address my comments and this manuscript should be published in EMBO J

Referee #3:

The revised manuscript by Tomomatsu et al., has addressed, as far as I can tell, not only all of my comments but also of those of all reviewers. It is an important study, of excellent quality in a topic of interest for a large public.

There was just some misunderstanding about one of my comments, and still a few language issues and a couple of typos, that could/should be corrected.

Description of these minor comments:

- 1) page 3, line 54, replace RPS20 by either Rps20 or encoded by RPS20
- 2) page 4, line 99conjugate to the various.....
- 3) page 5, line 110including the Ubi-Crest assay and the Ubi-uS10....
- 4) page 5, line 115containing the K48-linkage of the
- 5) page 6, line 139.....between the GFP and HIS open reading frames (ORF)....
- 6) page 6, lines 147 and 148..... (Fig. 1B), in good correlation with the accumulating polyubiquitin chain of uS10 in the ubp2 Δ and ubp3 Δ deletion mutants. (remove "could play a negative role in RQC induction". The idea comes later better supported in the text, see page 7, lines 164 and 165). Then start line 150 with "We next evaluated...." Instead of "we further confirmed..."
- 7) page 7, lines 157-159, "Ubp3 is involved in several mechanisms...., and the ubp3 Δ deletion mutant displayed a higher..... (I would remove the "since" and "could")
- 8) page 8, line 187, ...ubiquitin chains.....ubp2 Δ and ubp3 Δ mutants.....
- 9) in the text EV3D comes before EV3C, inverse the figure numbering

Point by point response to Reviewers (Submission ID: EMBOJ-2025-121138R)

We thank all reviewers for their positive, helpful, and insightful comments. In our detailed response, editorial issues and the reviewers' comments are italicized whereas our response is in Roman typeface with blue color.

Editorial issues:

- Please carefully go through the reference list and make sure that each reference is complete with citation year, volume, and page/locator numbers - this information is currently missing for several of them.

We have added the missing information (page/locator numbers) to the references where errors were found.

- When referring to the Appendix tables in the text, please make sure to maintain the correct naming ("Appendix Table S1/2").

We have corrected the naming of Appendix Table S1/2.

- Please check once more through the "Methods" section - e.g. in the first section, there is no reference(s) for "as previously described". Furthermore, in several instances μL or μM are wrongly written with a "u" instead of the Greek "Mu" symbol. And in places where you should have reused passages from previous papers word-by-word, please make sure to directly refer to the respective articles appropriately.

All instances of "ul" have been corrected to " μl ".

- Finally, please provide suggestions for a short 'blurb' text prefacing and summing up the study in two sentences (max. 250 characters), followed by 3-5 one-sentence 'bullet points' with brief factual statements of key results of the paper; they will form the basis of an editor-written 'Synopsis' accompanying the online version of the article. Please also upload a synopsis image, which can be used as a "visual title" for the synopsis section of your paper. The image should be in PNG or JPG format with the modest dimensions of EXACTLY 550 pixels wide and between 300 and 600 pixels high.

We have provided the blurb and bullet points in a separate document, and the synopsis image as a .jpg file.

Referee #1:

The authors have adequately addressed my comments and this manuscript should be published in EMBO J

We thank Referee #1 for their positive, helpful, and insightful comments during revision steps.

Referee #3:

The revised manuscript by Tomomatsu et al., has addressed, as far as I can tell, not only all of my comments but also of those of all reviewers. It is an important study, of excellent quality in a topic of interest for a large public.

We thank Referee #3 for their positive, helpful, and insightful comments during revision steps.

There was just some misunderstanding about one of my comments, and still a few language issues and a couple of typos, that could/should be corrected.

Description of these minor comments:

- 1) page 3, line 54, replace RPS20 by either Rps20 or encoded by RPS20
- 2) page 4, line 99conjugate to the various.....
- 3) page 5, line 110including the Ubi-Crest assay and the Ubi-uS10....
- 4) page 5, line 115containing the K48-linkage of the
- 5) page 6, line 139.....between the GFP and HIS open reading frames (ORF)....
- 6) page 6, lines 147 and 148..... (Fig. 1B), in good correlation with the accumulating polyubiquitin chain of uS10 in the *ubp2Δ* and *ubp3Δ* deletion mutants. (remove "could play a negative role in RQC induction". The idea comes later better supported in the text, see page 7, lines 164 and 165). Then start line 150 with "We next evaluated...." Instead of "we further confirmed..."

7)page 7, lines 157-159, "*Ubp3 is involved in several mechanisms....., and the ubp3Δ deletion mutant displayed a higher..... (I would remove the "since" and "could")*

8)page 8, line 187, *...ubiquitin chains.....ubp2Δ and ubp3Δ mutants.....*

9) *in the text EV3D comes before EV3C, inverse the figure numbering*

We appreciate all of Referee 3's comments and have made the necessary corrections accordingly.

Prof. Toshifumi Inada
The University of Tokyo
Division of RNA and gene regulation Institute of Medical Science,
4-6-1 shirokanedai
Minato-Ku, Tokyo 108-8639
Japan

4th Sep 2025

Re: EMBOJ-2025-121138R1
Polyubiquitin architecture editing on collided ribosomes maintains persistent RQC activity

Dear Prof. Inada,

Thank you for submitting your final revised manuscript for our consideration. I am pleased to inform you that we have now accepted it for publication in The EMBO Journal.

Yours sincerely,

Hartmut Vodermaier
